# Ce-induced synergistic effect in exsolved perovskite catalyst for highly efficient and robust methane dry reforming

Chencun Hao[1], Zhiyu Qu[2], Louise R. Smith [3], Nicholas F. Dummer [3], Haifeng Qi [3], Thomas J. A. Slater [3], Zhiping Zhu[4], Riguang Zhang [2]✉, Zhao Sun[1,3]✉, Zhiqiang Sun [1]✉ & Graham J. Hutchings [3]✉

Dry reforming of methane is an effective approach to convert two major greenhouse gases, methane and carbon dioxide, into high-value syngas, used as a feedstock for bulk and fine chemical synthesis. However, catalyst deactivation and carbon deposition under harsh conditions hinder its industrialization process. Herein, we present a Ce-modified and Ni-exsolved perovskite catalyst, $0.2Ce\text{-}La_{0.97}Ni_{0.4}Cr_{0.6}O_3$, for achieving highly efficient and robust $CH_4\text{-}CO_2$ reforming with $CH_4$ and $CO_2$ conversions of 87.4% and 92.9% at 800 °C, respectively. Moreover, this unique catalyst exhibits remarkable stability, maintaining its superior activity over 800 h. Characterization and density functional theory reveal that two Ce species are present: surface oxygen vacancy-moderate $CeO_{2-x}$ ($Ce_{surf}$) and bulk lattice Ce ($Ce_{bulk}$). These play a specific role in methane dry reforming, where the $Ce_{surf}$ promotes $CO_2$ adsorption and hinders carbon deposition, while $Ce_{bulk}$ induces lattice strain and Ni exsolution, key factors contributing to the high activity and stability.

Dry reforming of methane (DRM) is a critical process for simultaneously transforming two greenhouse gases, methane ($CH_4$) and carbon dioxide ($CO_2$), into syngas, a mixture of $H_2$ and CO with a near-equimolar $H_2$/CO ratio[1,2]. The generated syngas is a crucial feedstock for producing chemicals such as methanol and $C_{2+}$ via Fischer-Tropsch synthesis, and it can be further processed into high-value chemicals such as methanol and dimethyl ether[3,4]. Consequently, DRM enables both the efficient use of $C_1$ resources and the reduction of carbon emissions. However, the DRM process is constrained by the stability and energy consumption required to break the strong C-H (439.3 kJ mol⁻¹) and C=O (750 kJ mol⁻¹) bonds, posing significant operational challenges[5].

Catalysts play a pivotal role in lowering reforming temperature and energy consumption in DRM. Until now, significant progress has been made with non-noble metal-supported catalysts, such as Ni/ MgO[6], Ni/MgFeAlO₄[7], Ni/LaZrO₂[8], Co-La/Mg-Al₂O₃[9], Ni@Co/CeO₂[10], Co-Mg/TiO₂-Al₂O₃[11], Ni₃Fe₁/Al₂O₃[12], 0.3%Fe-Ni/Al₂O₃[13]. Nevertheless, to achieve high activity while simultaneously suppressing catalyst deactivation over the above-mentioned Fe-, Co-, and Ni-based catalysts remains a considerable challenge. Moreover, the occurrence of side reactions, including carbon deposition and reverse water-gas shift (rWGS), also lowers the H atom utilization efficiency and the overall efficiency of the DRM process.

Perovskite oxides, with their thermally stable crystal structures, have been extensively studied for catalytic applications at elevated temperatures. Particularly, exsolved perovskite materials, with B-site doping, exhibit excellent oxygen transport properties, with highly dispersed and reactive metal nanoparticles anchored on their surfaces that resist sintering[14]. Doping of B-sites in ABO₃-type perovskites with transition metals that exhibit low segregation energies promotes these

[1]Hunan Engineering Research Center of Clean and Low-Carbon Energy Technology, School of Energy Science and Engineering, Central South University, Changsha, China. [2]State Key Laboratory of Clean and Efficient Coal Utilization, College of Chemical Engineering and Technology, Taiyuan University of Technology, Taiyuan, China. [3]Max Planck-Cardiff Centre on the Fundamentals of Heterogeneous Catalysis FUNCAT, Cardiff Catalysis Institute, School of Chemistry, Cardiff University, Cardiff, UK. [4]State Key Laboratory of Coal Conversion, Institute of Engineering Thermophysics, Chinese Academy of Sciences, Beijing, China. ✉e-mail: zhangriguang@tyut.edu.cn; zhaosun@csu.edu.cn; zqsun@csu.edu.cn; Hutch@cardiff.ac.uk

metals to exsolve from the bulk lattice and anchor on the perovskite surface under reduction conditions[15,16]. This exsolution process generates reactive metal nanoparticles with strong metal-support interactions, which is to the benefit of sustaining catalytic activity[17–19]. Moreover, doping-induced lattice distortion would enhance the oxygen mobility of perovskite materials. This promotes the generation of more oxygen vacancies, thereby hindering carbon deposition during DRM due to the promotion of $O^{2-}_{lattice}$ transformation and the generation of surface O* species from $CO_2$ dissociation[20].

The A-site cations in perovskites, such as calcium[21], strontium[22], barium[23], lanthanum[24,25], cerium[26], praseodymium[27,28], neodymium[29], and others, typically do not directly participate in catalysis, but influence oxygen vacancy content and modulate the electronic properties of the B-site cations through charge balance control. Research has primarily focused on perovskites doped with B-site elements, including nickel[30], cobalt[31], and iron[32], as well as noble metals like ruthenium[33], rhodium[34], platinum[35], and iridium[36]. Nevertheless, further investigation is required to elucidate the regulation of particle size and density distribution of B-site transition metal exsolved nanoparticles, as well as the impact of A-site doping on the catalytic activity.

In this study, we developed a series of Ce-modified and Ni-doped $La_{0.97}Ni_{0.4}Cr_{0.6}O_3$ perovskite catalysts, with the Ni exsolved ($Ni_{ex}$) $La_{0.97}Ce_{0.03}Ni_{0.4}Cr_{0.6}O_3$ catalysts obtained after hydrogen reduction. Cerium oxides ($CeO_{2-x}$), due to their $Ce^{3+}/Ce^{4+}$ redox pair, are a potential oxygen storage-release material for DRM[27,37,38]. Additionally, as far as we are aware, few studies have explored the impact of Ce modification on the exsolution and the catalytic properties of perovskites. Our study revealed that the Ce species exist in two distinctive forms within the perovskite catalyst: A-site lattice Ce, which induces lattice distortion and promotes Ni exsolution, and surface oxygen vacancy ($O_v$)-abundant Ce species ($CeO_{2-x}$), which provides oxygen storage and a release cycle microenvironment, which inhibits carbon accumulation and promotes $CO_2$ activation. Here, we demonstrate the synergistic effect of these two forms of Ce in enhancing DRM activity and stability.

## Results and discussion
### Catalyst performance
The DRM performance of $Ni_{ex}$-$La_{0.97}Ce_{0.03}Ni_{0.4}Cr_{0.6}O_3$ catalyst with varying Ce doping ratios is shown in Fig. 1a. With increasing Ce ratio from 0 to 0.2, $CH_4$ and $CO_2$ conversion rates improved to 91.1% and

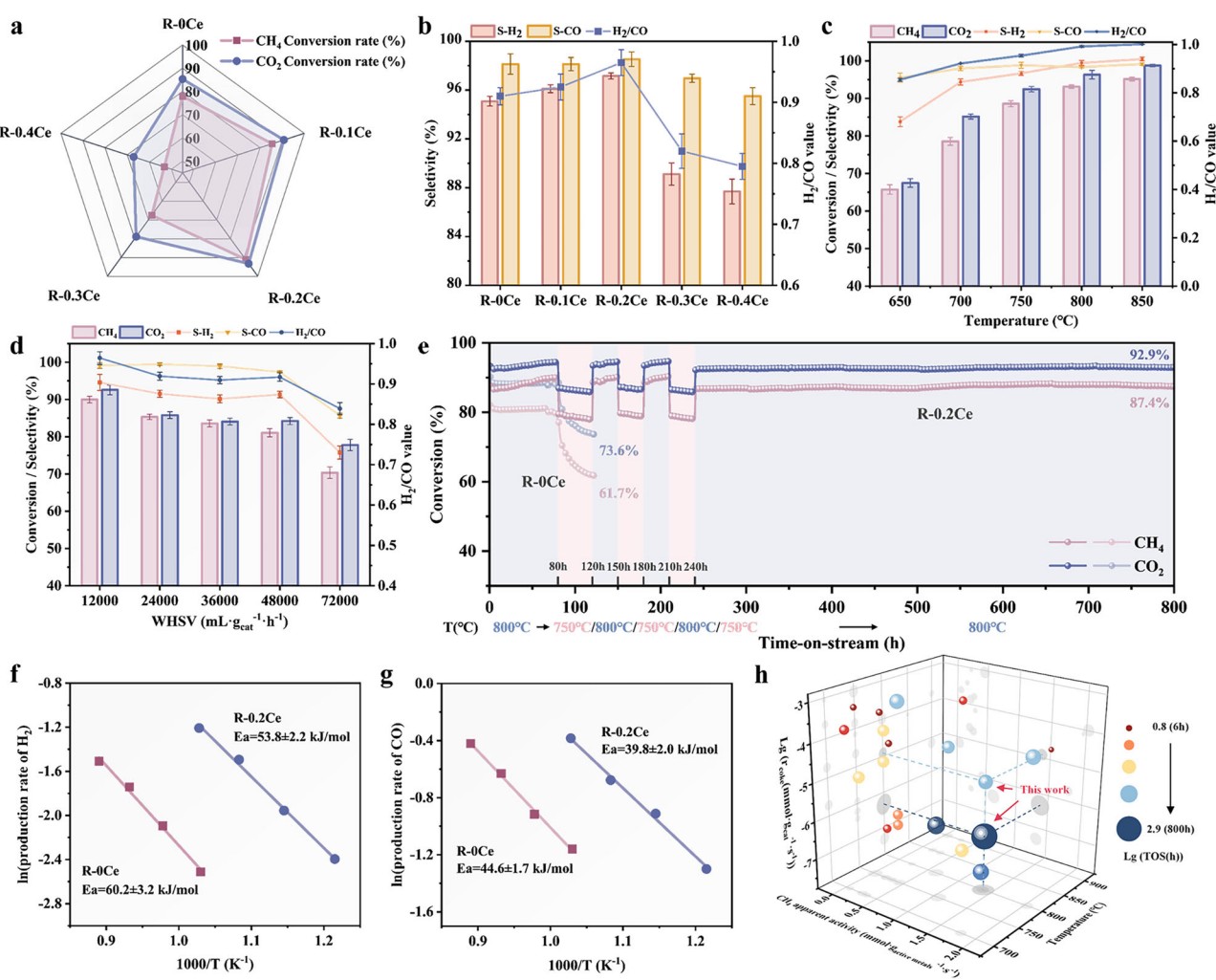

**Fig. 1 | DRM performance of R-xCe samples. a** $CH_4$, $CO_2$ conversion rates over various samples with different Ce doping ratios in the DRM reaction. **b** $H_2$, CO selectivity and $H_2$/CO ratio. **c** The conversion rates of $CH_4$ and $CO_2$, selectivity of $H_2$ and CO, and $H_2$/CO ratio at different temperatures for sample 0.2Ce. **d** The conversion rates of $CH_4$ and $CO_2$, selectivity of $H_2$ and CO, and $H_2$/CO value at different WHSV for sample 0.2Ce at 750 °C. **e** Stability test of R-0.2Ce and R-0Ce for 800 h and 120 h. Evaluated conditions: 800/750 °C, $CH_4$: $CO_2$: $N_2$ = 1: 1: 1, WHSV = 30,000 mL g$^{-1}$ h$^{-1}$. Arrhenius plot in terms of the rate of product of **f** $H_2$ and **g** CO catalyzed by the R-0Ce and R-0.2Ce catalysts. 25 mg of each catalyst was loaded into the quartz tube. **h** Comparison of the DRM performance with literature (Table S1).

93.3%, respectively. However, excessive Ce (R-0.4Ce) resulted in a sharp decline in the respective conversions to 54.0% and 67.2%. This trend reflects a balance between optimal Ce doping and excessive coverage of Ni-active sites by $CeO_{2-x}$. $H_2$ and CO selectivity, as well as the $H_2/CO$ ratio also exhibit a similar volcanic trend with increasing Ce doping (Fig. 1b). Reactions at different temperatures (650–850 °C) over R-0.2Ce demonstrated a significant increase of $H_2$ (83.8% → 96.7%), CO (95.6% → 98.8%) selectivity, and the $H_2/CO$ ratio (0.85 → 0.95) from 650 to 750 °C, while the increase gradually slows down at a temperature > 750 °C (Figs. 1c and S1–S3). This phenomenon can be explained by the occurrence of rWGS ($H_2 + CO_2 \rightarrow CO + H_2O$, $\Delta H = +41$ kJ/mol) equilibrium, and Ce doping modulates its dynamic equilibrium as well as acts to increase the H utilization efficiency (Figs. S4 and S5). The $CH_4$ conversion of 95.2% and $CO_2$ conversion of 98.8% with a $H_2/CO$ ratio of 1.0 were achieved at 850 °C, a relatively greater performance in comparison to previously reported examples.

The influence of different weight hourly space velocities (WHSVs) on DRM performance remained comparable across WHSVs from 12,000 to 48,000 mL $g^{-1} h^{-1}$. High $CH_4$ and $CO_2$ conversions of 81.1% and 84.2%, respectively, were maintained below a WHSV value of 48,000 mL $g^{-1} h^{-1}$ (Fig. 1d). The optimal R-0.2Ce catalyst was subjected to long-term stability measurements, which showed abrupt and obvious deactivation at the initial stage of the DRM reaction, and then recovered, possibly due to surface reconstruction in a dynamic equilibrium between $CH_4$ and $CO_2$ (Figs. S6–S11). Furthermore, the R-0.2Ce catalyst was subjected to 800 h long-term stability test under more severe operating conditions (Figs. 1e and S12). It demonstrates excellent stability, maintaining a $CH_4$ conversion rate of 87.4% and a $CO_2$ conversion rate of 92.9% after 800 h, with negligible deactivation compared to its initial performance. In contrast, the R-0Ce sample exhibits a gradually declining performance after ~65 h, with pronounced deactivation observed after switching to 750 °C at 80 h. This resulted in reduced $CH_4$ and $CO_2$ conversion rates of 61.7% and 73.6%, respectively.

As shown in Fig. 1f, g, the apparent activation energies for $H_2$ and CO yields exhibited differences in the rate control steps for the same samples, suggesting the distinct rate-determining steps for $H_2$ and CO production. Furthermore, R-0.2Ce demonstrated lower apparent activation energies compared to R-0Ce, indicating the enhancement in Ce modification. This phenomenon may be attributed to an increase in the interfacial redox capacity via $CeO_x \leftrightarrow CeO_{x-\delta}$ looping according to the following characterizations. The R-0.2Ce catalyst exhibits higher apparent methane activity and lower carbon formation rate at the same DRM temperatures, which demonstrates obvious cutting-edge compared with previously reported catalysts (Fig. 1h and Table S1).

## Structural characterizations

Powder XRD patterns of the calcined perovskite oxides, $LaCrO_3$, $LaNi_{0.4}Cr_{0.6}O_3$, and $CeO_{2-x}$-$La_{0.97}Ce_{0.03}Ni_{0.4}Cr_{0.6}O_3$, all display a similar perovskite phase containing $Cr^{6+}$ ($La_2CrO_6$) after B-site Ni doping (Fig. S13, and see Table S2 for their ICP-OES results). We consider that this arose from maintaining the charge balance of the bulk phase. (TEM images of C-0Ce and C-0.2Ce as shown in Figs. S14 and S15). In the R-0.2Ce sample, a peak belonging to metallic Ni(111) is observed at 44.5°, while the main perovskite peak at 32.6° shifts to a lower angle compared with the calcined one (Fig. S16). This implies that part of the B-site Ni has exsolved from the bulk phase to form $Ni^0$ nanoparticles on the surface.

Transmission electron microscopy (TEM) images of the R-0.2Ce sample reduced at 900 °C confirmed Ni nanoparticle exsolution (Fig. 2a). Additionally, energy-dispersive X-ray spectroscopy (EDS) mapping confirms the homogeneous distribution of La, Ce, Cr, and O elements, evidencing the formation of the La-Ce-Ni-Cr-O perovskite. The Ni nanoparticle, exposed (111) crystal plane, was proportionally embedded within the parent perovskite, according to the HR-TEM

(Fig. 2b). AFM further confirmed the exsolved Ni nanoparticles distributed on the perovskite surface (Fig. S17). In addition, a partially reduced $CeO_{1.75}(220)$ species was also observed on the perovskite surface layer.

The effect of Ce doping on the particle size and distribution of exsolved structure was investigated (Fig. 2c–e). Among the investigated samples, R-0.2Ce possessed the smallest average particle size and the highest dispersion of exsolved Ni nanoparticles. It was demonstrated that Ce doping is conducive to the exsolution of Ni nanoparticles when compared with R-0Ce, possibly because Ce doping into the A-site lattice induces lattice distortion and reduces the energy barrier for Ni exsolution. Nevertheless, excessive Ce doping leads to significant increase in the average particle diameters (~95.8 nm) of the exsolved Ni (Fig. S18). Excessive Ce modification in the R-0.4Ce sample leads to $CeO_{2-x}$ aggregation on the perovskite surface, which hinders Ni segregation and aggregates the nanoparticles on the surface, resulting in a significant decrease in the density of active sites and an increase in the particle size.

Prior research has demonstrated that the reduction temperature exerts a pronounced influence on the dimensions and distribution of the exsolved nanoparticles[39,40]. The LaMer model proposed by Sugimoto et al.[41] and the findings from O'Leary et al.[42] suggest that high-temperature reduction increases Ni segregation and transformation rates. Nevertheless, the growth barrier is meanwhile reduced, which results in a decrease in the density and an increase in the size of the exsolved Ni particles, leading to less dispersed nanoparticles. Therefore, we examined the evolution of surface morphology with the R-0.2Ce sample as a function of reduction temperature from 700 to 1000 °C (Figs. S19–S22). The results are in accordance with the previous work, i.e., high reduction temperatures promote Ni exsolution but also cause nanoparticle aggregation, while low temperatures inhibit the exsolution process. The DRM test further confirmed that the sample reduced at 900 °C exhibited the highest catalytic performance, attributed to its smaller particle sizes and higher dispersion (Figs. S23–S25). Specifically, this sample achieved the highest particle density of ~9.3 NPs/μm², with an average particle diameter of 37.9 nm. The DRM test also demonstrated the best performance over the sample reduced at 900 °C (Fig. S26).

Using aberration-corrected high-angle annular dark-field scanning transmission electron microscopy (AC-HAADF-STEM) combined with EDS mapping (Fig. 2f), we confirmed Ce doping at the A-site and Ni doping at the B-site of the $LaCrO_3$ perovskite. Additionally, a proportion of Ce was enriched as a secondary phase on the perovskite surface. EDS mapping of La/Ni shows non-overlapping distributions, testifying to their occupations of A- and B-sites, respectively. Some Ni undergoes segregation and aggregates in the central region (Fig. S27). Comparison of La/Ni and La/Cr mappings reveals that Ni occupies the B-site perovskite together with Cr. Ce/Cr mappings further reveal the co-existence of two Ce species: Ce doped into the A-site and thin $CeO_{2-x}$ nanolayers present on the surface and bulk. The measured lattice parameters ($a = 0.558$ nm, $c = 0.572$ nm) are consistent with those derived from XRD refinement (Pnma space group). The atomic intensity signals within the orange boxes were analyzed, confirming the doping of Ce at the A-site of the perovskite structure (Fig. 2g, h).

To further clarify the effect of Ce doping on the Ni exsolution, XRD Rietveld refinement of C-0Ce and C-0.2Ce samples was carried out. Results confirm the Pnma space group (Figs. S28 and S29) and a reduced cell volume when compared to $LaCrO_3$, confirming partial replacement of B-site Cr by smaller-radius Ni (Tables S3 and S4). Ce doping also reduces the cell volume further (233.92 Å³ → 233.58 Å³), as $Ce^{3+}$ (0.134 nm) and $Ce^{4+}$ (0.114 nm) have smaller ionic radii than $La^{3+}$ (0.136 nm)[26]. The $LaNi_{0.4}Cr_{0.6}O_3$ perovskite yielded a tolerance factor of 0.97, indicating the presence of lattice distortion after B-site Ni doping, see detailed introduction in supplementary information.

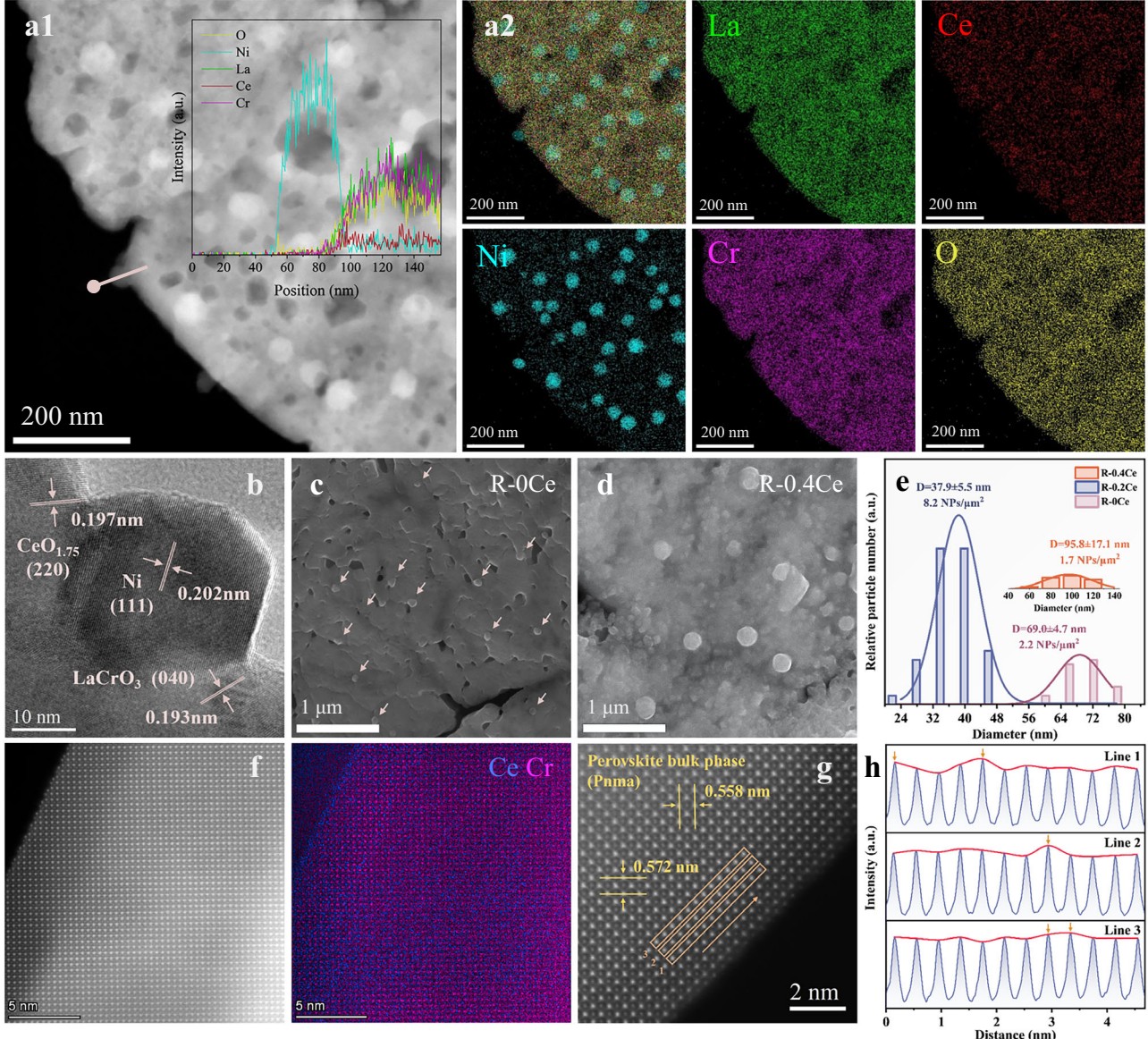

**Fig. 2 | Microstructure and morphology studies of R-xCe samples. a1** HAADF-STEM image of R-0.2Ce. **a2** EDS mapping of La, Ce, Ni, Cr, and O elements. **b** HR-SEM image of the exsolved Ni nanoparticle of R-0.2Ce. **c** SEM images of R-0Ce. **d** SEM images of R-0.4Ce. **e** Particle size distribution of exsolved nanoparticles of three samples (R-0Ce, R-0.2Ce, R-0.4Ce). **f** AC-HAADF-STEM images of R-0.2Ce and EDS mapping of Ce and Cr elements. **g** AC-HAADF-STEM image of R-0.2Ce and **h** corresponding showcasing diagram of perovskite A-site content in three layers.

After reduction, the average cell volumes of both R-0Ce and R-0.2Ce remain smaller than that of LaCrO₃, representing successful substitution of Cr by Ni due to its smaller radius. The reduced samples were fitted by Pnma and R-3c space groups (Tables S3 and S4). The exsolved Ni nanoparticles of R-0Ce and R-0.2Ce account for 3.51 wt% and 6.79 wt%, respectively, resulting in a 93.4% promoted Ni exsolution after Ce doping. Furthermore, the average cell volume of R-0.2Ce (282.44 Å³) is obviously smaller than that of R-0Ce (309.41 Å³), further demonstrating the enhancement of Ni exsolution under Ce doping[43]. Furthermore, the increased percentage of asymmetry orthorhombic (Pnma) structures[44] after Ce doping (59.2% for R-0.2Ce and 35.9% for R-0Ce) indicates greater lattice distortion, which facilitates Ni exsolution (Figs. 3a, b and S30).

X-ray absorption spectroscopy (XAS) characterization was further conducted to determine the chemical and electronic states of Ni and Ce. We observed the presence of a shoulder at the high-energy flank of the white line (8.353 keV) and the plateau period between 8.357 keV and 8.363 keV for the normalized Ni K-edge X-ray absorption near edge structure (XANES) of the R-0Ce and R-0.2Ce samples, which can be regarded as distinctive signatures of Ni incorporation into the lattice of perovskite, confirming the substitution of Ni (Fig. 3c)[45,46]. The absorption threshold $E_0$ of both samples is situated between Ni and NiO, indicating that their average valence states are between 0 and +2. The slightly lower $E_0$ energy of R-0.2Ce than that of R-0Ce further suggests the formation of more embedded Ni⁰ exsolved nanoparticles in R-0.2Ce.

Fourier-transformed R-space from Extended X-ray absorption fine structure (EXAFS) spectra of R-0Ce and R-0.2Ce further confirms the lattice distortion caused by B-site Ni doping. According to the curve fitting of EXAFS data, the R-0Ce and R-0.2Ce possess two forms of Ni-O bonds, Ni-O₁ and Ni-O₂ paths, which correspond to bond lengths of 1.89 and 2.02 Å, and 1.92 and 2.06 Å, respectively (Figs. 3d, e, and S31, Table S5). The prolonged Ni-O bond lengths after Ce doping point to its role in enhancing the lattice oxygen mobility, along with the two longer Ni-O paths of R-0.2Ce, which show lower coordination numbers, also support the existence of more oxygen vacancies. Further,

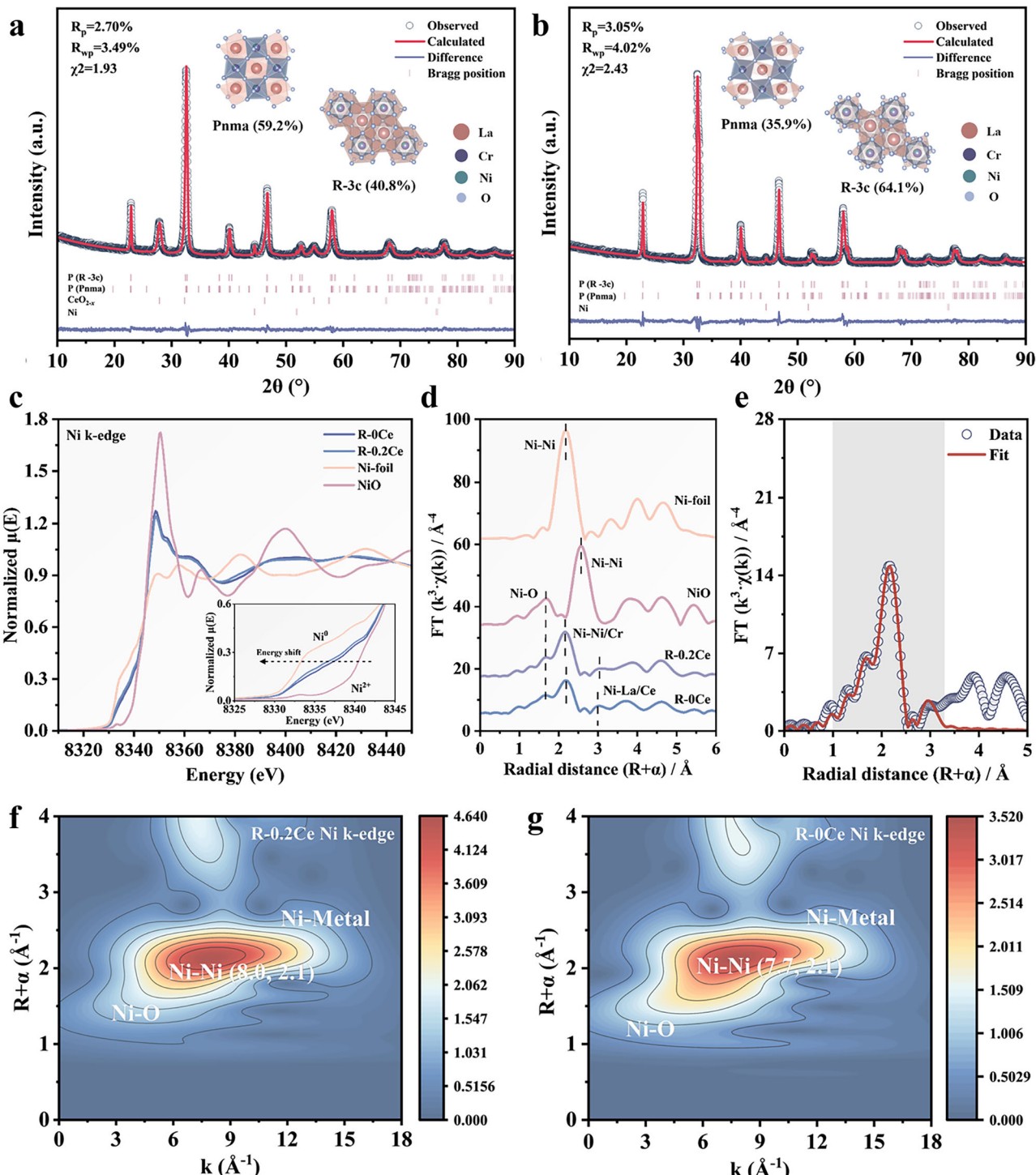

**Fig. 3 | Fine-structure characterizations of R-0Ce and R-0.2Ce samples. a** XRD refinement data of R-0.2Ce. **b** XRD refinement data of R-0Ce. **c** Normalized Ni K-edge XANES data of the R-0.2Ce and R-0Ce. **d** Ni K-edge EXAFS data of R-0.2Ce and R-0Ce in *R*-space. **e** Fourier transform of the $k^3$-weighted EXAFS curves (empty dots) and fit model (red line) for the reduced R-0.2Ce samples. The light gray shadow indicates the fitted region. **f** R-0.2Ce $k^3$-weighted wavelet transform plots of Ni K-edge EXAFS spectra. **g** R-0Ce $k^3$-weighted wavelet transform plots of Ni K-edge EXAFS spectra.

the R-0.2Ce sample shows a stronger Ni-Ni signal (at ~2.49 Å), which implies the promotion of B-site Ni exsolution after doping with Ce. Results from XANES linear combination fitting corroborate this speculation (Fig. S32 and Table S6). The scattering peak of the second shell layer is assigned to Ni-La or Ni-Ce paths. The bond lengths of R-0.2Ce (3.28 Å) are slightly shorter than those of R-0Ce (3.33 Å), contributing to the A-site substitution of La by Ce.

To differentiate the backscattering atoms, wavelet transform analysis was conducted (Figs. 3f, g, S33 and S34). The intensity centers in k-space (strongest oscillation) of both R-0Ce and R-0.2Ce are between Ni foil (8.3 Å$^{-1}$) and Ni-O coordination (6.1 Å$^{-1}$), revealing the possible existence of Ni-Cr coordination[47]. Combined with XRD and XAS results, two Ce species, $CeO_{2-x}$ with abundant oxygen vacancies and lattice Ce within the A-site perovskite structure, were therefore

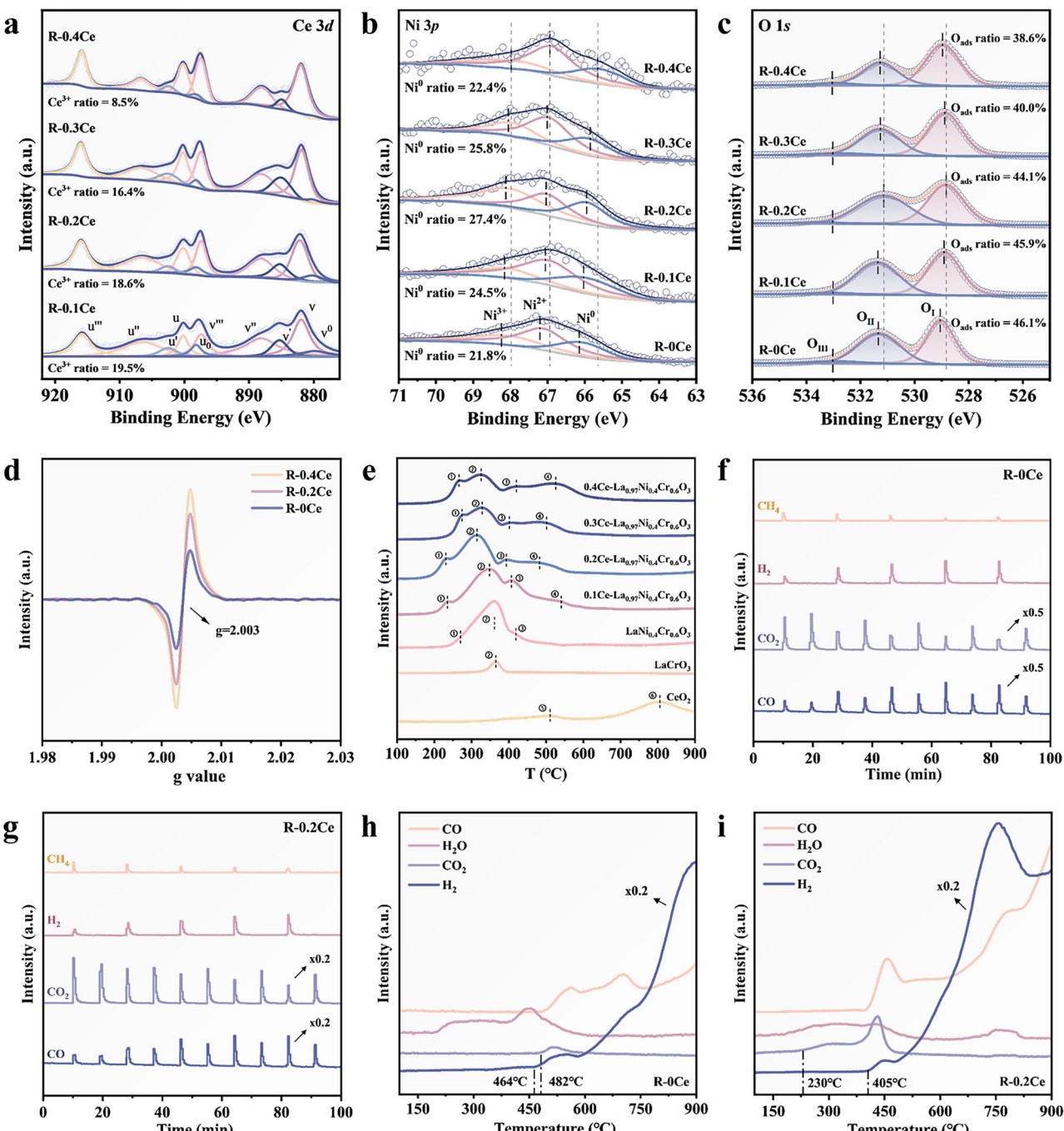

**Fig. 4 | Characterizations of $x$Ce-La$_{0.97}$Ni$_{0.4}$Cr$_{0.6}$O$_3$ perovskite catalysts.** XPS spectra of **a** Ce 3$d$. **b** Ni 3$p$. **c** O 1$s$. **d** The EPR spectra of R-0Ce, R-0.2Ce, and R-0.4Ce. **e** H$_2$-TPR profiles of calcined catalysts. MS signal of CH$_4$/CO$_2$ alternately pulse experiments at 750 °C on **f** R-0Ce and **g** R-0.2Ce. CH$_4$-TPR of **h** R-0Ce, and **i** R-0.2Ce.

confirmed. (Other relevant XAS data can be found in Figs. S35 and S36).

XPS analysis demonstrated a decrease in the Cr$^{3+}$/(Cr$^{3+}$+Cr$^{4+}$) ratio upon Ce doping, suggesting that the Ce$^{4+}$ may promote the conversion of Cr$^{3+}$ to Cr$^{4+}$ through redox balance for its high oxidation capacity. (Figs. 4a−c and S37, Table S7). This alters the charge balance and could lead to electron accumulation for B-site cations[48]. The peak binding energy shifts of B-site elements in perovskite structures reflect variations in B−O bond strength[49]. Results from Cr 2$p$ and Ni 3$p$ indicate the gradual increase and decrease in the binding energy with increasing amounts of Ce introduced, suggesting a strengthening of the Cr-O bond and a weakening of the Ni-O bond. This also indicates that the metallic Ni is in an electron-rich state, and the electron cloud density

can be modulated by the regulation of the Ce doping amount. The shift of the O 1$s$ binding energy decreases first and then increases, which reflects an average B-O bond strength with Ce doping, as confirmed by the Raman spectroscopy (Fig. S38). The surface Ce$^{3+}$ content in the R-0.4Ce sample decreased to 8.5%, likely due to the aggregation of surface CeO$_{2-x}$ species at higher Ce concentrations, which favors the formation of more stable Ce$^{4+}$ species. Additionally, the XPS signal intensity of metallic Ni$^0$ was significantly reduced in the R-0.4Ce sample, while the Ni$^{2+}$ peak was notably enhanced. These results suggest that the excess CeO$_{2-x}$ phase promotes the chemical oxidation of surface Ni while physically hinders the exposure of metallic Ni$^0$ through surface coverage. As confirmed by TEM results, it is revealed that excessively high Ce doping is not conducive to exsolving Ni

nanoparticles. XPS under different atmospheres was applied to elucidate the alterations in the electronic structure and oxidation state at the catalyst interface in methane and carbon dioxide atmospheres, as shown in Figs. S39–S41, Table S8.

EPR results also supported the enhancement in oxygen vacancy formation with elevated Ce doping, and the amount of the oxygen vacancies formed follows the order of R-0.4Ce > R-0.2Ce > R-0Ce (Fig. 4d). Combined with XPS results, we conclude that a moderate surface and bulk oxygen vacancy distribution promises optimal carbon resistance and $CO_2$ activation performance. $H_2$-TPR tests were carried out to examine the reducibility of the perovskite catalysts (Fig. 4e). $H_2$-TPR profiles of four Ce-modified $La_{0.97}Ni_{0.4}Cr_{0.6}O_3$ samples correspond to the transformation of $Ni^{3+} \rightarrow Ni^{2+}$, $Cr^{6+} \rightarrow Cr^{4+/3+}$, $Ni^{2+} \rightarrow Ni^0$, and $Ce^{4+} \rightarrow Ce^{3+}$[26]. Among them, R-0.2Ce exhibits the lowest preliminary reduction temperature of ~240 °C, confirming its exceptional lattice oxygen mobility.

Alternate $CH_4$-$CO_2$ pulse experiments were conducted to reveal the capability of R-$x$Ce in C-H bond activation and lattice oxygen mobility at 750 °C (Figs. 4f, g and S42). Among the three tested samples, R-0.4Ce shows inferior performance due to the suppression of Ni exsolution by the highly oxidized state of the surface and the coverage of active sites by $CeO_{2-x}$ species, resulting in a low methane conversion; while over R-0Ce a much higher $H_2$/CO ratio was achieved, demonstrating the influence of increased carbon deposition. We anticipate that the superiority of R-0.2Ce is due to: (i) high lattice oxygen mobility supplied by surface $CeO_{2-x}$ species and bulk-phase perovskite. This hinders carbon deposition with abundant oxygen vacancy generated; (ii) high $CO_2$ activation and splitting capability. This is actuated by the formed oxygen vacancies, thus inducing sustainable $O^{2-}_{lattice}$ release-supply looping.

To further confirm the impact of lattice oxygen mobility of exsolved perovskite on methane activation and conversion, $CH_4$-TPR tests were carried out (Figs. 4h, i and S43). The R-0.2Ce and R-0.4Ce samples detected an increased $CO_2$ signal at temperatures as low as 230 °C and 247 °C, respectively, demonstrating their capability in low-temperature $CH_4$ oxidation and enhancement in lattice oxygen mobility. $H_2$ signals emerged subsequently at 405 °C, accompanied by CO evolution. In contrast, over R-0Ce, $H_2$ signals were recorded at 464 °C, followed by $CO_2$ and CO signals at approximately 482 °C, corresponding to much weaker peak intensities. Notably, the $H_2$ peaks observed over the Ce-doped samples reached their highest value at approximately 750 °C, followed by a decline and subsequent rise. This phenomenon could be attributed to the reduction of a substantial quantity of bulk-phase oxygen in $CeO_{2-x}$ at elevated temperatures ($H_2$-TPR of $CeO_2$), which then combines with $H_2$ to generate $H_2O$. This process also results in the attenuation of the methane partial oxidation reaction due to the decreased $H_2$ and CO signals. The CO signal intensity of the R-0Ce samples is markedly lower than that of the Ce-containing samples, which can be assigned to the lower surface oxygen vacancies formed. In summary, two forms of Ce species, surface $O_v^-$-abundant $CeO_{2-x}$ and A-site lattice Ce, concurrently contribute to highly efficient $CH_4$ oxidation, $CO_2$ dissociation, and exsolution of Ni nanoparticles, thereby enhancing $CH_4$-$CO_2$ reforming with a highly efficient and robust catalyst.

The structural, morphological, and surface properties of the as-prepared R-0Ce and R-0.2Ce samples and after long-term DRM testing were characterized by XRD, XPS, Raman, etc. Powder XRD results indicate that the perovskite phase and exsolved Ni remain basically unchanged after long-term DRM test, demonstrating its stability (Figs. 5a and S44). XPS spectra of reacted R-0.2Ce for Ce 3$d$, O 1$s$, Ni 3$p$, and Cr 2$p$ are displayed in Fig. 5b–e, which demonstrated a notable increase in the $Ce^{3+}$ species. $Ce^{3+}$ exhibits strong redox capabilities that promote the dynamic capture and release of lattice oxygen, thereby facilitating the removal of surface carbon. The observed increase in $Ni^0$ content is attributed to the reducing environment of DRM, which

facilitates the continuous segregation of Ni ions from the perovskite bulk phase, forming a larger population of $Ni^0$ species. This dynamic migration contributes to the sustained enhancement in catalytic activity during the initial reaction stage, reflecting a dynamic equilibrium within the catalytic system under DRM conditions.

Thermogravimetric analysis (Fig. 5f) revealed that the carbon deposition on R-0.2Ce after 800 h of reaction led to only a 1.6% weight loss, while R-0Ce showed a 6.4% loss after just 120 h, underscoring the superior carbon resistance of R-0.2Ce. Raman spectroscopy further confirmed the reduced carbon deposition after Ce doping (Fig. S45).

SEM and TEM characterizations (Figs. S46 and S47) showed that the post-reaction morphology of R-0.2Ce closely resembles that of the fresh sample, with only minimal formation of carbon nanotubes. In contrast, R-0Ce displayed significant sintering of the perovskite surface and agglomeration of Ni nanoparticles, along with more pronounced carbon nanotube formation. Quantitatively, the average Ni particle size in R-0.2Ce increased moderately from 37.9 nm to 61.8 nm after 800 h, whereas in R-0Ce it grew substantially from 69.0 nm to 116.9 nm. These results confirm that Ce doping not only mitigates carbon deposition but also improves the anti-sintering stability of the catalyst.

To gain insights into the intermediate species and the reaction pathways during the DRM process, in-situ DRIFTS experiments were conducted (Figs. 6a, b and S48). Introducing 10 vol% $CH_4$ into the reactor at 500 °C resulted in a pronounced $CH_4$ peak at approximately 3016 $cm^{-1}$, while peaks representing CO (~2200 $cm^{-1}$) and $CO_2$ (~2360 $cm^{-1}$) emerged due to the partial and complete oxidation of $CH_4$[50]. The weak CO peak observed on the R-0Ce sample aligns with the preceding $CH_4$-TPR results, indicating its limited capacity to activate $CH_4$ at 500 °C. Additionally, two bands corresponding to the deformation vibrations of $CH_x^*$ ($x \leq 2$) and $CH_3^*$ were observed at ~1335 $cm^{-1}$ and ~1350 $cm^{-1}$[51,52], resulting from the dissociation of C-H bonds at the three-phase interface of $Ni_{ex}$, $CeO_{2-x}$, and $La_{0.97}Ce_{0.03}Ni_{0.4}Cr_{0.6}O_3$ with the assistance of lattice oxygen activation. Upon introducing 10 vol% $CO_2$, the vibration of the hydroxyl group (OH*) was detected at ~3600-3800 $cm^{-1}$[53], while the prominent peak at ~2360 $cm^{-1}$ was attributed to the gaseous $CO_2$. Oxygen vacancies adsorb and dissociate $CO_2$ to produce O* species, leading to the generation of bidentate carbonates, HCOO*, and $CH_xO^*$ at ~1580 $cm^{-1}$, 1550 $cm^{-1}$, and 1390 $cm^{-1}$, respectively[54–57]. Notably, the R-0Ce sample exhibited a weaker peak for $CH_xO$, elucidating a relatively low capability for $CO_2$ activation. As the atmosphere shifted to 10 vol% $CH_4$-10 vol% $CO_2$, the $CH_x^*$ vibration signal (~1330 $cm^{-1}$) was significantly weakened and slightly shifted to a higher wavelength, corresponding to the $CH_3^*$ vibration (~1350 $cm^{-1}$). This suggests that the catalysts we designed effectively hinder methane decomposition during the DRM process, leading to the combination of $CH_x$ with O* or H* with $CO_x^*$ to produce key oxygen-containing intermediate species[28], such as $CH_xO^*$, HCOO*, OH*, and bidentate carbonates.

Figure 7a shows that the segregation energy required for Ni exsolved from the B-site of C-0.2Ce catalyst decreases significantly with increasing Ce doping. When Ce is doped into the first layer, the segregation energy decreases by 7.3%, from −1.23 eV to −1.32 eV. When Ce is doped into the third layer, the segregation energy decreases from −1.23 eV to −2.36 eV. This indicates that A-site Ce doping significantly promotes the in-situ exsolution of Ni, which well explains the phenomenon of greater density distribution of surface Ni nanoparticles and enhanced DRM activity in our experiments.

Our experimental characterization confirms that $CH_4$ is initially activated through C-H bond cleavage to form $CH_3^*$, which is then dehydrogenated to a $CH_2^*$ intermediate[54]. From this point, two potential reaction pathways were investigated: oxidation of $CH_2^*$ to $CH_2O^*$, and further dehydrogenation of $CH_2^*$ to CH* (Figs. 7b and S49, S50). On the R-0Ce catalyst, $CH_2O^*$ formation has a high activation free energy of 2.00 eV and a reaction free energy of 0.58 eV,

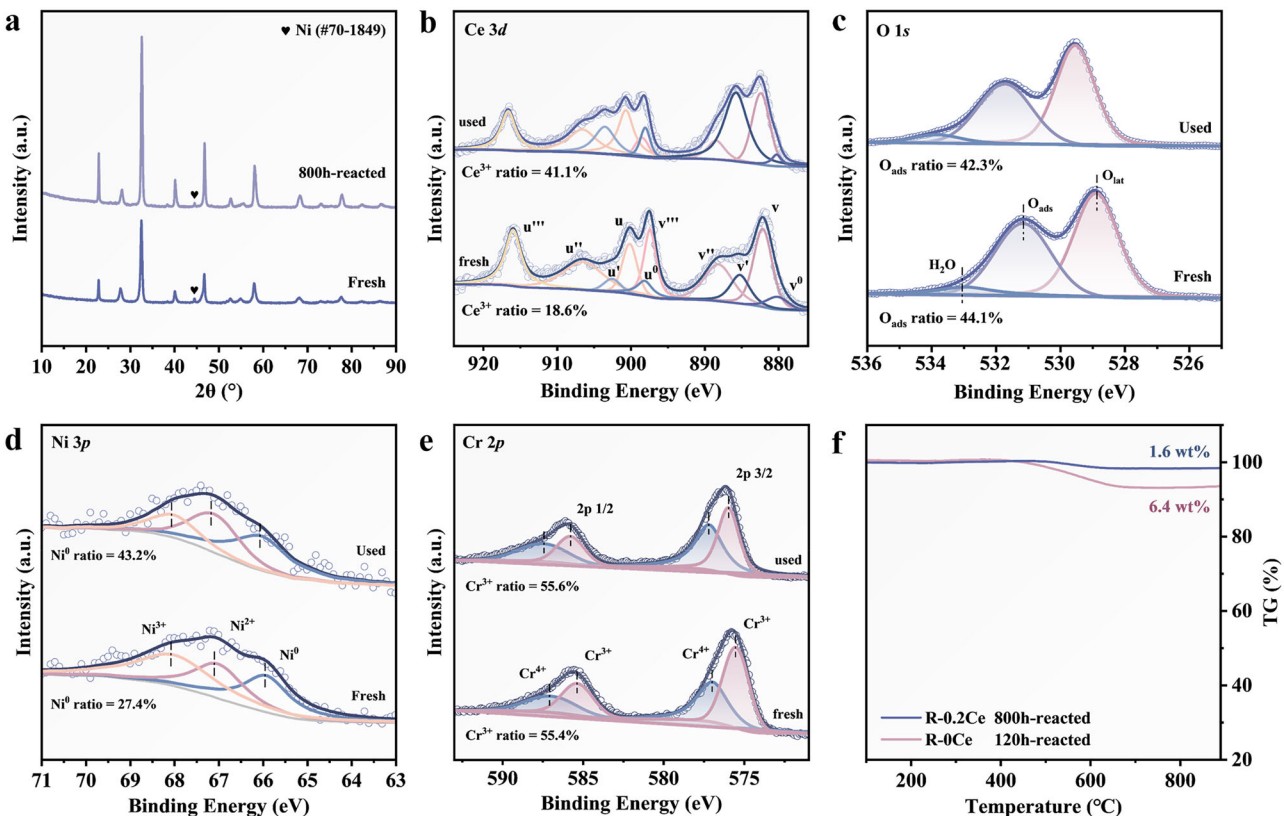

**Fig. 5 | Characterizations of long-term reacted $x$Ce-La$_{0.97}$Ni$_{0.4}$Cr$_{0.6}$O$_3$ perovskite catalysts. a** XRD pattern of the reacted samples. XPS spectra of **b** Ce 3$d$. **c** O 1$s$. **d** Ni 3$p$. **e** Cr 2$p$. **f** TG of 120h-reacted R-0Ce and 800 h-reacted R-0.2Ce.

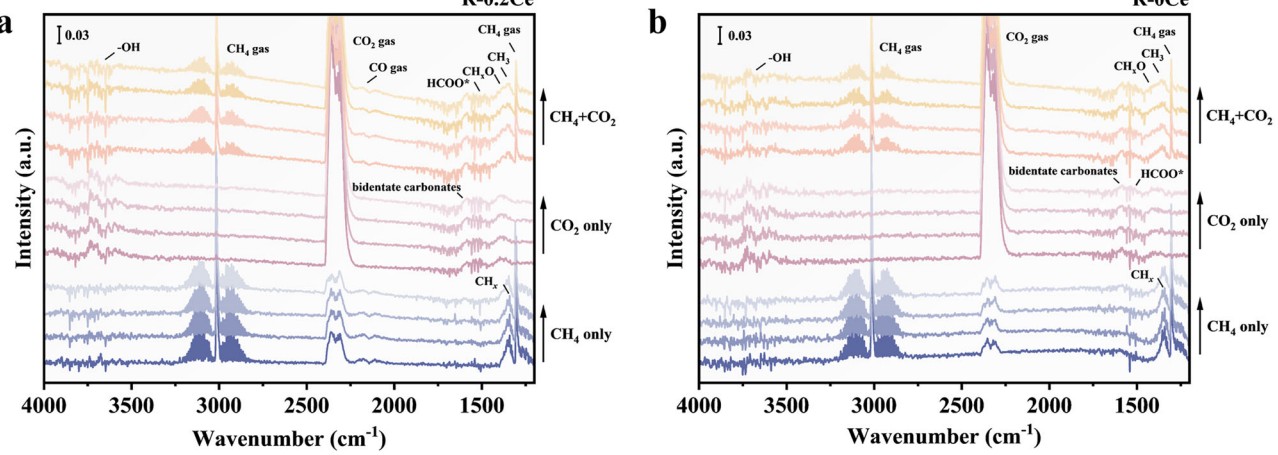

**Fig. 6 | In situ DRIFTS spectra. a** R-0.2Ce and **b** R-0Ce at 500 °C, expose to 10% CH$_4$ for 20 min, 10% CO$_2$ for 20 min, and 10% CH$_4$-10% CO$_2$ for 20 min.

whereas CH$_2$* dehydrogenation to CH* is kinetically more favorable, with an activation free energy of 1.02 eV and a reaction free energy of 0.49 eV. Therefore, CH* formation dominates on R-0Ce, leading to a higher risk of carbon accumulation. In contrast, on the R-0.2Ce catalyst (Figs. 7c and S49, S50), CH$_2$* more readily reacts with active oxygen to form CH$_2$O*, with a lower activation free energy of 1.57 eV, while the competing dehydrogenation pathway to CH* requires 1.74 eV. This suggests that Ce doping suppresses excessive CH$_2$* dehydrogenation, thus inhibiting carbon deposition and promoting CH$_2$O* formation. Additionally, the oxidation of CH$_2$* to CH$_2$O* on R-0.2Ce is more kinetically favorable compared to R-0Ce (1.57 eV vs. 2.00 eV), further confirming the role of Ce in mitigating carbon buildup.

As shown in Fig. 7d, e, the $\varepsilon_d$ value of R-0.2Ce (−1.84 eV) is closer to the Fermi level than that of R-0Ce (-2.03 eV), indicating stronger CO$_2$ adsorption and activation on R-0.2Ce. This facilitates C-O bond cleavage and the generation of adsorbed CO and reactive oxygen species. These species can effectively react with carbon derived from CH$_2$* dehydrogenation, thereby suppressing carbon accumulation. Moreover, differential charge density and Bader charge analysis (Fig. 7f, Tables S9 and S10) reveal electron transfer from the CeO$_{2-x}$ cluster to the Ni cluster, significantly enhancing the metal-support interaction. This results in stronger anchoring of exsolved Ni nanoparticles to the perovskite surface, improving their thermal stability and reducing sintering during DRM, as confirmed by SEM and TEM analyses. The electron donation also increases the $d$-orbital electron density of Ni,

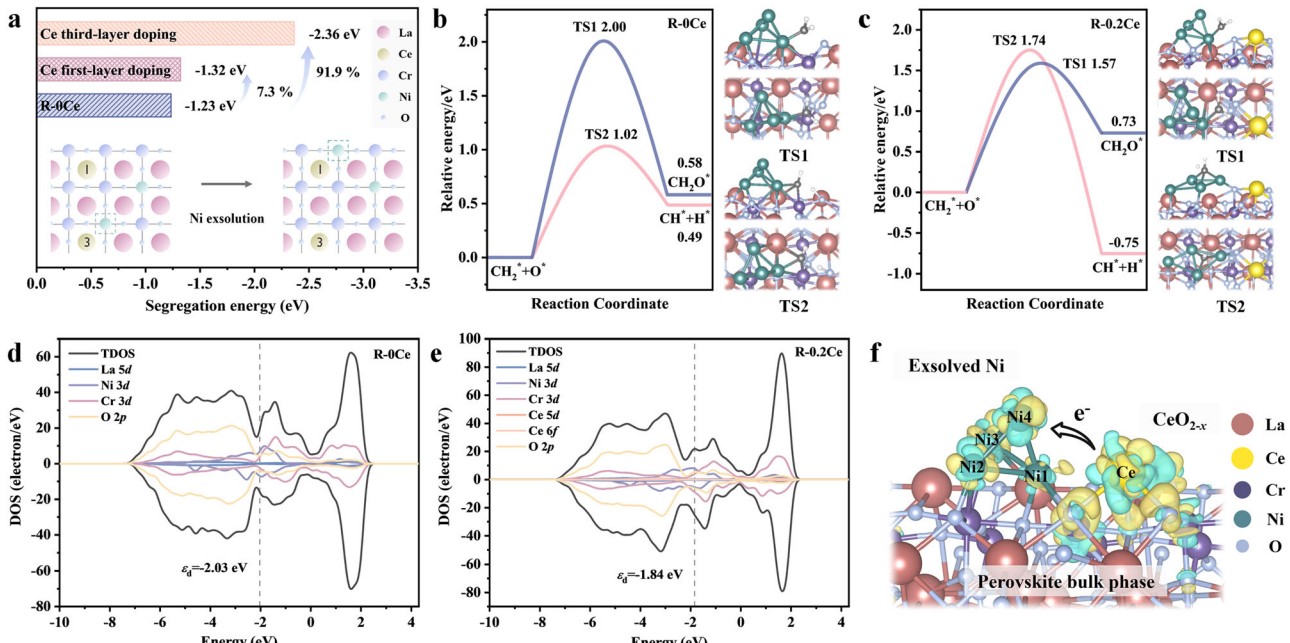

**Fig. 7 | DFT calculations. a** (111) oriented slab model and the segregation energy defined as the energy change upon swapping Ni and Cr between the first and third layers. Free energy profiles for the reactions related to CH$_2^*$ intermediate on the **b** R-0Ce and **c** R-0.2Ce catalysts. Density of states and *d*-band center for **d** R-0Ce and **e** R-0.2Ce catalyst surfaces. **f** The differential charge density of R-0.2Ce catalyst. Cyan and yellow regions present the electron accumulation and depletion, respectively. La, Ce, Cr, Ni, and O atoms are shown in red, yellow, purple, green, and blue, respectively.

potentially boosting the adsorption and activation of both CH$_4$ and CO$_2$.

In summary, we report a Ce-modified and Ni-exsoluted Ni$_{ex}$-La$_{0.97}$Ni$_{0.4}$Cr$_{0.6}$O$_3$ perovskite catalysts that demonstrates superior DRM activity and satisfactory stability. The 0.2Ce-La$_{0.97}$Ni$_{0.4}$Cr$_{0.6}$O$_3$ catalyst exhibits exceptional CH$_4$ and CO$_2$ conversions of 87.4% and 92.9%, respectively, along with a desirable H$_2$/CO ratio of 0.93, over 800 h at 800 °C. Characterizations including XRD refinement, AC-HADDF-STEM, XPS, XAS, etc. confirmed that the two Ce species exploited a synergistic effect on DRM activity promotion. Firstly, the A-site bulk lattice Ce induces lattice distortion, reducing the segregation energy required for the exsolution of Ni at the B-site of perovskite. It promotes the ability of the catalyst to activate C-H bonds. Secondly, surface oxygen vacancy-moderate Ce species (CeO$_{2-x}$) exists beyond the perovskite surface and acts as a looping carrier, enhancing CO$_2$ activation and suppressing carbon deposition. The modulation of two distinct forms of Ce offers novel insights for designing high-performance exsoluted perovskite catalysts with enhanced DRM efficiency.

## Methods
### Catalyst synthesis
All the chemicals used in this study were received from Sinopharm Chemical Reagent Co. Ltd. The catalysts were synthesized by the Pechini method[58]. Five samples, LaNi$_{0.4}$Cr$_{0.6}$O$_3$, $x$Ce-La$_{0.97}$Ni$_{0.4}$Cr$_{0.6}$O$_3$ ($x$ = 0.1, 0.2, 0.3, or 0.4), were synthesized according to different Ce:La:Ni:Cr ratios of $x$: 0.97: 0.4: 0.6. In a typical synthesis, stoichiometric amounts of lanthanum nitrate hexahydrate (La(NO$_3$)$_3$ 6H$_2$O), cerium nitrate hexahydrate (Ce(NO$_3$)$_3$ 6H$_2$O), nickel nitrate hexahydrate (Ni(NO$_3$)$_2$ 6H$_2$O), chromic nitrate (Cr(NO$_3$)$_3$ 9H$_2$O) were dissolved in deionized water to obtain a solution of 0.25 M. After that, citric acid and ethylene glycol were added into the solution with a mole ratio of metal ions (Ce$^{4+}$, La$^{3+}$, Ni$^{2+}$, and Cr$^{3+}$): citric acid: ethylene glycol equal to 1.0: 1.5: 3.0. Then, the solution was stirred in a water bath at 80 °C until a gel was formed. The formed gel was then dried in a drying oven at 150 °C for 12 h, and the obtained solid was subsequently ground to fine

powder within a particle size range of 0.15 ~ 0.20 mm. The collected powder was calcined in a muffle furnace in air for 6 h at 850 °C with a heating rate of 2 °C/min. Finally, the calcined sample was reduced by 10 vol% H$_2$ for 3 h under various temperatures (700, 800, 900, or 1000 °C). The samples $x$Ce-La$_{0.97}$Ni$_{0.4}$Cr$_{0.6}$O$_3$ samples before and after reduction are abbreviated as C-$x$Ce and R-$x$Ce, respectively (see Table S11 for detailed List of Abbreviations).

### Catalyst characterization
The contents of La, Ce, Ni, and Cr in each sample were measured by inductively coupled plasma optical emission spectroscopy (ICP-OES, PerkinElmer Avio-500).

The crystal structures of the samples were performed by an X-ray diffractometer (PANalytical Empyren) using Cu Kα (λ = 1.5406 Å) radiation 45 kV and 40 mA, operating on a continuous scan mode. The X-ray patterns was recorded in a range of 5-90° at a scanning rate of 5°/min. XRD refinement data were processed by a Fullprof software[59].

Scanning electron microscopy (SEM, TESCAN MIRA4 LMH) with an energy-dispersive X-ray spectroscopy detector (EDS mapping) to characterize the morphology of exsoluted perovskite samples and analyze the distribution of Ce, La, Ni, Cr, and O elements.

FEI Tecnai G2 F20 TEM was used to characterize the surface morphology of the catalyst and the element distribution. The samples used for TEM measurements were put into ethanol to obtain a suspension (1 mg mL$^{-1}$) and then treated by ultrasonic for 5 min.

High-angle annular dark-field scanning transmission electron microscopy (HAADF-STEM) was performed using a JEOL JEM-ARM200F instrument.

Bruker Dimension edge Atomic Force Microscope (AFM) was used to explore the surface morphology and the roughness of the R-0.2Ce catalysts.

X-ray photoelectron spectroscopy (XPS) measurements were performed on a Thermo Scientific TM K-Alpha TM+ spectrometer equipped with a monochromatic Al Kα X-ray source (1486.6 eV) to

analyze the chemical states of the catalysts. The binding energies were calibrated by taking C1*s* peak (284.8 eV) as a reference.

Raman measurements were conducted using a Renishaw inVia Raman spectrometer with a 532 nm laser wavelength. Powder samples were applied to a glass slide and flattened for consistency. Spectra were acquired from various positions on each sample, ensuring that each sample underwent a minimum of three tests.

The detection of oxygen vacancies was performed using electron paramagnetic resonance (EPR) spectroscopy, tested on BRUKER EMXPLUS EPR in continuous wave electron paramagnetic resonance.

In-situ diffuse reflectance infrared Fourier transform spectroscopy (In-situ DRIFTS) was conducted by using a Nicolet iS50 spectrometer equipped with an In-situ diffuse reflectance cell and a mercury-cadmium-telluride detector. In a typical measurement, the perovskite catalyst was initially heated from room temperature to 500 °C with a heating rate of 10 °C/min and maintained for 30 min under $N_2$ (30 mL/min), and the background spectrum was collected after stabilization. Afterwards, 10 vol% $CH_4$ in $N_2$, 10 vol% $CO_2$ in $N_2$, and a mixture of 10 vol% $CH_4$ and 10 vol% $CO_2$ in $N_2$ were successively introduced into the cell for 20 min. The spectra were collected every 5 minutes, which lasted for 60 minutes in total.

The temperature-programmed reaction of methane ($CH_4$-TPR) measurements were performed by using a chemisorption analyzer (Micromeritics AutoChem II 2920) connected with a mass spectrometer (MS, Hiden-20 R&D). In a typical experiment, the sample was placed into the U-type quartz reactor and heated at 300 °C for 30 min under He (20 mL/min) with a heating rate of 10 °C/min. The sample was then cooled to room temperature, followed by the introduction of 10 vol% $CH_4$ in He (20 mL/min). Meanwhile, MS is conducted to record the $CH_4$ signal. Once the $CH_4$ curve is stabilized, the reactor was heated from room temperature to 900 °C with a heating rate of 10 °C/min. The outlet products, $H_2$, $CH_4$, $H_2O$, CO, and $CO_2$, were detected by MS according to the m/z results of 2, 16, 18, 28, and 44.

The $CH_4$-$CO_2$ transient alternating pulse measurements were conducted by using a chemisorption analyzer (Micromeritics Auto-Chem II 2920). In a typical measurement, 100 mg sample was put into the reactor and pretreated by He (20 mL/min) for 30 min at 300 °C with a heating rate of 20 °C/min. Then, the $CH_4$-$CO_2$ transient alternating pulse experiments were performed at 750 °C by switching the reaction atmosphere between 10 vol% $CH_4$ in He and 10 vol% $CO_2$ in He for every 7 min. The outlet products were also detected by MS.

$H_2$-TPR measurements were conducted by using the same chemisorption analyzer. In a typical test, 100 mg calcined sample was pretreated in Ar (40 mL/min) at 300 °C for 30 min, and the temperature was then reduced to 50 °C. Once the baseline stabilized, the temperature was increased from 50 to 900 °C with a heating rate of 10 °C/min, meanwhile switching the atmosphere to 10 vol% $H_2$ in Ar (40 mL/min). The hydrogen consumption signals were detected using a thermal conductivity detector.

The XAS (Ni K-edge) were collected at BL14W beamline in Shanghai Synchrotron Radiation Facility (SSRF). The storage rings of SSRF were operated at 3.5 GeV with a stable current of 200 mA. Using Si(111) double-crystal monochromator, the data collection was carried out in fluorescence mode using a Lytle detector. All spectra were collected in ambient conditions. XAS data were analyzed using the Demeter software package[60]. A linear function was subtracted from the pre-edge region, then the edge jump was normalized using the Athena software. The χ(k) data were isolated by subtracting a smooth, third-order polynomial approximating the absorption background of an isolated atom. The k3-weighted χ(k) data were Fourier transformed after applying a Hanning window function (Δ$k$ = 1.0). The global amplitude EXAFS (CN, R, $σ^2$ and Δ$E_O$) were obtained by nonlinear fitting, with least-squares refinement, of the EXAFS equation to the Fourier-transformed data in R-space, using Artemis software.

## Catalytic performance

Experimental tests were performed in a fixed-bed reactor (600 mm in length and 8 mm in inner diameter). In a typical experiment, 300 ± 5 mg (60 mg for long-term test) of reduced catalyst was placed into the isothermal temperature interval of a quartz tube, as supported by an appropriate amount of quartz wool. Before each test, the sample-loaded quartz tube was purged with 30 mL/min $N_2$ for 10 minutes. Then, the furnace is heated from room temperature to the target temperature (650, 700, 750, 800, or 850 °C). Once the temperature reaches the target one and stabilizes, a total flow rate of 25 vol% $CH_4$ and 25 vol% $CO_2$ in $N_2$ was introduced ($CH_4$: $CO_2$: $N_2$ = 1: 1: 1 for long-term test). The long-term experimental test was conducted for 800 h. The outlet products, $H_2$, CO, $CO_2$, and $CH_4$, were determined by a GC (INFICON Micro GC Fusion).

The DRM performances were evaluated by $CH_4$, $CO_2$ conversion rates, $H_2$, CO selectivity, and $H_2$/CO mole ratio. Total flow rate of the outlet gas ($F_{total}$, mol s$^{-1}$) is defined as:

$$F_{total} = \frac{F_{in}(N_2)}{C_{out}(N_2)} \tag{1}$$

where $C_{out}(N_2)$ refers to the outlet concentration of $N_2$, $F_{in}(y)$ represents the feeding rate of $y$, $y = N_2$, $CH_4$, or $CO_2$. The outlet flow rate of gas $i$ ($F_{out}(i)$, mol s$^{-1}$) is calculated as:

$$F_{out}(i) = F_{total} \times C_{out}(i) \tag{2}$$

where $i$ refers to possible components of $CH_4$, $CO_2$, CO, or $H_2$. The conversion rates of $CH_4$ and $CO_2$, $X(CH_4, \%)$ and $X(CO_2, \%)$, were calculated as:

$$X_{CH_4} = \frac{F_{in}(CH_4) - F_{out}(CH_4)}{F_{in}(CH_4)} \times 100\% \tag{3}$$

$$X_{CO_2} = \frac{F_{in}(CO_2) - F_{out}(CO_2)}{F_{in}(CO_2)} \times 100\% \tag{4}$$

The $H_2$/CO ratio were calculate by the follow equations:

$$H_2/CO = \frac{F_{out}(H_2)}{F_{out}(CO)} \tag{5}$$

The error bars in Figs. 1, S1–S3, and S26 represent the standard deviation (SD) obtained from three independent performance tests.

## DFT calculation

Density functional theory (DFT) calculations are implemented in the Vienna ab initio Simulation Package (VASP 5.4.4)[61]. Perdew-Burke-Ernzerhof with generalized gradient approximation (GGA) was used to describe the exchange-correlation functional[62]. A vacuum layer of 15 Å was introduced to avoid interactions between periodic images. A cutoff energy of 400 eV for the plane-wave basis sets and a 3 × 3 × 1 grid generated by the Monkhorst-Pack method were adopted for geometric optimization calculations[63]. Meanwhile, the k-point grid in the Brillouin zone was set to be 8×8×1 by the Monkhorst-Pack scheme when calculating the electronic structure. Structure optimizations were performed until the total energies converged to $5 \times 10^{-6}$ eV and the forces acting on the relaxed ions were less than 0.05 eV/Å. To counteract erroneous electron delocalization, GGA + U methods were applied to La, Cr and Ni atoms. Values of $U_{La} = 5$ eV, $U_{Cr} = 2$ eV, $U_{Ni} = 3.5$ eV, $U_{Ce} = 6$ eV were utilized[64–67]. The climbing-image nudged elastic band (CI-NEB) method[68] was employed to search for the transition states (TS), then, the dimer method was used to optimize the located TS[69]. Moreover, the TS structure was confirmed with only one imaginary frequency.

Both R-0Ce and R-0.2Ce catalyst models were constructed based on the experimental characterization results that the (111) crystal plane was the dominantly exposed crystallographic surface. As shown in Fig. S50, The R-0Ce catalyst was modeled using a three-layer $p(2 \times 1)$ $LaCrO_3$(111) supercell, in which the bottom two layers were fixed. In this model, a $Ni_4$ cluster was located on the surface, with an adjacent oxygen vacancy surrounding the $Ni_4$ cluster. The R-0.2Ce model was obtained by loading a $CeO_2$ cluster on the R-0Ce surface and creating an oxygen vacancy within the cluster by removing one oxygen atom, resulting in a $CeO_{2-x}$ cluster structure. To calculate Ni segregation energy on the (111) plane, a model for Ni-doped $LaCrO_3$ catalyst was constructed using a five-layer $p(1 \times 1)$ supercell with the bottom two layers fixed.

The adsorption free energy ($G_{ads}$), activation free energy ($G_a$), and reaction free energy ($\Delta G$) were determined by the following equations[70]:

$$G_{ads}(T, P) = E_{mol/surface} - E_{mol} - E_{surface} + \Delta U_{ads} + \Delta E_{ZPE_{ads}} - T\Delta S_{ads} \tag{6}$$

$$G_a(T, P) = E_{TS} - E_{IS} + \Delta E_{ZPE_a} + \Delta U_a - T\Delta S_a \tag{7}$$

$$\Delta G(T, P) = E_{FS} - E_{IS} + \Delta E_{ZPE} + \Delta U - T\Delta S \tag{8}$$

where $E_{mol/surface}$, $E_{mol}$, and $E_{surface}$ were the total free energies of the adsorbed system, the clean surfaces, and the free molecules, respectively. Considering our optimal catalytic reaction conditions were 750 °C and 1.0 atm, all energies were corrected at 1023.15 K and 1.0 atm using VASPKIT software.

## Data availability

All original data needed to evaluate the conclusions in the paper have already been present in the manuscript and the Supplementary Information (including Supplementary Figs. S1–S50 and Tables S1–S11). Data are available from the corresponding authors upon request.

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

## Acknowledgements

Z.S. acknowledges the support of the National Natural Science Foundation of China (52476144, 42441835). G.J.H. acknowledges the support of RCUK|Engineering and Physical Sciences Research Council (EPSRC)–EP/W014408/1.

## Author contributions

Z.S., Z.Q.S., and G.J.H. supervised this research and conceived the idea. C.C.H. and Z.S. executed the perovskite preparation and characterization measurements, analyzed the experimental data, and wrote the original manuscript. The DFT analysis was assisted by Z.Y.Q. and R.G.Z. The manuscript was revised by C.C.H., Z.Y.Q., L.R.S., N.F.D., H.F.Q., T.J.A.S., Z.P.Z., R.G.Z., Z.S., Z.Q.S., and G.J.H. All authors discussed the results and commented on the paper.

## Competing interests

The authors declare no competing interests.
