## [Transparent Peer Review file · Nature Communications]

Ce-induced synergistic effect in exsolved perovskite catalyst for highly efficient and robust dry reforming of methane

Corresponding Author: Professor Graham Hutchings

Version 0:

Reviewer comments:

Reviewer #1

(Remarks to the Author)

I hereby report on the manuscript by Hao et al. on Ce-induced synergistic effect in exsolved perovskite catalyst for highly efficient and robust dry reforming of methane. While I acknowledge that the scientific work has been carried out without flaws and the topic itself is for sure timely, I recommend rejection of the manuscript in the present form for the following reason:

- 1) I am not convinced that the conclusions drawn from the results are either spectacularly new or are unambiguously connected to the results. Specifically, the role of oxygen vacancies in DRM has been pointed out multiple times. Secondly, the authors claim that Ce(bulk) induces lattice strain and Ni exsolution - however, there is no direct experimental proof for that. This is a pity, because such methods (4D electron microscopy) exist.
- 2) In addition, there are multiple sentences in the manuscript that make no sense. For example, on page 14 the sentence in line 271 starting with "As confirmed by..." The word "conductive" is wrong here.

Reviewer #2

(Remarks to the Author)

The paper you mentioned provides valuable insights into the effects of Ce doping on dry methane reforming. However, the stability test duration of 120 hours is significantly shorter compared to other studies, making the claims of stability questionable. Additionally, using dilute gases like nitrogen at levels exceeding 50% during DRM measurements does not seem appropriate for assessing the inherent performance of the catalyst. Several studies cited in the paper do not use dilute gases and report catalysts that maintain high performance and stability over 1000 hours. Therefore, it is recommended that experimental data obtained without inert gases be presented. To meet the standards of publication in Nature Communications, it is crucial to adequately address the comments mentioned above.

Reviewer #3

(Remarks to the Author)

This is a very strong paper with significant novelty and impact. The thorough characterization, mechanistic insights, and excellent performance of the catalyst make it a valuable contribution to the field of dry reforming of methane. The weaknesses are relatively minor and can be addressed with some clarifications and a slightly expanded discussion in certain areas. I would highly recommend this paper for publication after minor revisions addressing the points below.

- While the paper mentions carbon deposition and shows evidence of MWCNT formation, a more in-depth discussion of the type of carbon formed (beyond Raman ID/IG ratios) would be beneficial. TEM images after the long-term stability test are shown in the SI, showing MWCNTs. Elaborating on the mechanism of carbon resistance (why the MWCNTs don't lead to rapid deactivation in the R-0.2Ce sample) in the main text would strengthen the conclusions. The connection between the increased Ce³⁺-O_v structures after reaction (Fig 5b,c) and carbon resistance could be explicitly stated.
- While Figure 1h and Table S1 compare the DRM performance to previous research, a more direct comparison in the discussion section to similar perovskite-based catalysts (specifically those with Ni and/or Ce) would be helpful. This would more clearly highlight the advantages of the current catalyst design. Are there other examples of dual-role Ce doping in perovskites, and how does this work compare?

- The authors correctly note that the reduction temp can influence the final product. While the supplementary material explores this, a brief discussion of the optimized reduction temp (900C) in the main body is warranted. Just a sentence or two to guide the readers to the related figures.
- The long term stability test used a WHSV of 12000 mL·g⁻¹·h⁻¹. How does this compare to the space velocities where the performance remained constant? Does this low value contribute to high performance?
- While the H₂/CO ratio approaches the theoretical value of 1, it's consistently slightly below 1. A brief discussion of potential reasons for this (e.g., slight contribution of rWGS, even if minimized) would add completeness.
- The explanation for the decreased performance with excessive Ce (R-0.4Ce) could be strengthened. While the authors mention coverage of Ni active sites by CeO_{2-x}, explicitly connecting this to the XPS and TEM results (showing larger Ni particle size and potentially less exposed Ni) would be beneficial.
- The TG data (Fig. S44) shows more carbon deposition on R-0.2Ce, but this sample is more stable. The paper mentions a lower graphitization degree, but a clearer, more concise statement connecting this to the overall stability is needed.

Minor comments

- While generally very well-written, there are a few minor grammatical errors. A final proofread is recommended.
- It would improve the readability if the authors had listed all the abbreviations at the end of the paper.

Version 1:

Reviewer comments:

Reviewer #1

(Remarks to the Author)

The authors have significantly improved their work by incorporating the input from both referees and I therefore recommend acceptance of the manuscript.

Reviewer #3

(Remarks to the Author)

I would like to thank the authors for their detailed revisions. They have answered all my previous comments, and also the comments from the other reviewers. The paper is now much stronger and the quality is greatly improved.

I was very happy to see the new 800-hour stability test. This experiment was done under more difficult conditions (higher WHSV, less N₂), and it provides strong proof of the catalyst's stability. The analysis of the catalyst after the test is also very clear and answers my questions about carbon resistance.

The other points I raised are also now corrected. The text explains the choice of the reduction temperature, discusses the H₂/CO ratio, and the comparison to other works in Figure 1h is now clearer.

I also want to note the new data that was added in response to the other reviewers. The new AC-HAADF-STEM images and the DFT calculations are very important additions. They give direct support for the authors' main conclusions about the role of Cerium. This makes the scientific argument of the paper much more solid.

The revisions are very good and have significantly improved the manuscript.

I recommend that this paper should be accepted for publication.

A point-to-point response to the reviewer's comments

Reviewer #1:

I hereby report on the manuscript by Hao et al. on Ce-induced synergistic effect in exsolved perovskite catalyst for highly efficient and robust dry reforming of methane. While I acknowledge that the scientific work has been carried out without flaws and the topic itself is for sure timely, I recommend rejection of the manuscript in the present form for the following reason.

Our response: Thank you very much for your valuable feedback and constructive comments. We appreciate your thoughtful assessment of our manuscript and the opportunity to improve its clarity and scientific rigor. This paper focuses on the synergistic effect of A-site doped Ce and surface CeO_{2-x} species. The synergistic effect of the two species effectively enhances the activation of CH₄, CO₂, and significantly increases the anti-carbon-accumulation ability, resulting in a comprehensive optimization of the catalytic performance. Based on the reviewers' comments and advice, we have thoroughly revised our manuscript:

1. AC-HAADF-STEM images and EDS mapping analyses have been added, which provide direct evidence of Ce doping at the A-site, the presence of CeO_{2-x} species on the surface, as well as the segregation of Ni at the B-site of the perovskite.
2. DRM stability test with higher airspeed and lower N₂ dilution ratio was added, and the R-0.2Ce sample maintained 87.4% CH₄ and 92.9% CO₂ conversion after 800 h reaction.
3. Further TG analysis of the carbon deposition in the post-reaction samples showed that the coke formation rate of R-0.2Ce is only 4.63E-07 mmol·g_{cat}⁻¹·s⁻¹, while R-0Ce is 1.23E-05 mmol·g_{cat}⁻¹·s⁻¹.
4. We further complement the first-principles calculations by revealing that Ce doping at the A-site will effectively reduce the segregation bias energy of Ni at the B-site. In addition, DRM reaction energy barriers were calculated.

Comment 1: I am not convinced that the conclusions drawn from the results are either spectacularly new or are unambiguously connected to the results. Specifically, the role of oxygen vacancies in DRM has been pointed out multiple times. Secondly, the authors claim that Ce(bulk) induces lattice strain and Ni exsolution - however, there is no direct experimental proof for that. This is a pity, because such methods (4D electron microscopy) exist.

Our response: Thank you for your in-depth review of our paper. In this paper, we focus on the synergistic effects of dual Ce species: surface CeO_{2-x} inhibits carbon deposition and enhances catalytic activity by inhibiting sintering through electron transfer with exsolved Ni nanoparticles. The variable valence state ($\text{Ce}^{4+}/\text{Ce}^{3+}$) of CeO_{2-x} generates oxygen vacancies, and these reactive oxygen species oxidize the accumulated carbon. Meanwhile, CeO_{2-x} enhances CO_2 adsorption and dissociation, contributing to the removal of carbon species and sustained oxygen recharge. Whereas bulk-phase Ce doping induces lattice distortions, reduces the segregation energy of Ni in the lattice, and creates more anchored active sites, which enhances catalytic activity and stability. The catalyst showed excellent stability in a new 800 h long-term DRM test.

Detailed revisions: In response to your comments, we have thoroughly revised the main article to better emphasizing our innovative aspects, and prove our conclusions through more characterizations and theoretical calculations. Detailed revisions are provided below.

Line 171: Using aberration-corrected high-angle annular dark-field scanning transmission electron microscopy (AC-HAADF-STEM) combined with EDS mapping (Figs. 2f-2g), we confirmed Ce doping at the A-site and Ni doping at the B-site of the LaCrO_3 perovskite. Additionally, a proportion of Ce was enriched as a secondary phase on the perovskite surface. EDS mapping of La/Ni shows non-overlapping distributions, testifying their occupations of A- and B-sites, respectively. Some Ni undergoes segregation and aggregates in the central region (Fig. S27). Comparison of La/Ni and La/Cr mappings reveals that Ni occupies the B-site perovskite together with Cr. Ce/Cr mappings further reveal the co-existence of two Ce species: Ce doped into the A-site and thin CeO_{2-x} nanolayers present on the surface and bulk. The measured lattice

parameters ($a = 0.558$ nm, $c = 0.572$ nm) are consistent with those derived from XRD refinement (Pnma space group). The atomic intensity signals within the orange boxes were analyzed, confirming the successful doping of Ce at the A-site of the perovskite structure (Fig. 3g-3h).

Fig. 2 | Microstructure and morphology studies of R-xCe samples. a1 HAADF-STEM image of R-0.2Ce. **a2** EDS mapping of La, Ce, Ni, Cr, and O elements. **b** HR-SEM image of the exsolved Ni nanoparticle of R-0.2Ce. **c** SEM images of R-0Ce. **d** SEM images of R-0.4Ce. **e** Particle size distribution of exsolved nanoparticles of three samples (R-0Ce, R-0.2Ce, R-0.4Ce). **f** AC-HAADF-STEM images of R-0.2Ce and EDS mapping of Ce and Cr elements. **g** AC-HAADF-STEM image of R-0.2Ce and **h** corresponding showcasing diagram of perovskite A-site content in three layers.

Fig S27. AC-HAADF-STEM-EDS mapping of **a** La/Ni and **b** La/Cr elements.

Moreover, we have updated Fig. 7 in the main text with DFT calculations, including the B-site Ni segregation energy, activation energy barrier calculation, density of states and *d*-band center, and the charge density of catalysts.

Line 382: Fig. 7a shows that the segregation energy required for Ni exsolved from the B-site of C-0.2Ce catalyst decreases significantly with increasing Ce doping. When Ce is doped into the first layer, the segregation energy decreases by 7.3%, from -1.23 eV to -1.32 eV. When Ce is doped into the third layer, the segregation energy decreases from -1.23 eV to -2.36 eV. This indicates that A-site Ce doping significantly promotes the in-situ exsolution of Ni, which well explained the phenomenon of greater density distribution of surface Ni nanoparticles and enhanced DRM activity in our experiments.

Line 389: Our experimental characterization confirms that CH₄ is initially activated through C-H bond cleavage to form CH₃^{*}, which is then dehydrogenated to a CH₂^{*} intermediate ^[54]. From this point, two potential reaction pathways were investigated: oxidation of CH₂^{*} to CH₂O^{*}, and further dehydrogenation of CH₂^{*} to CH^{*} (Fig. 7b, Figs. S49-S50). On the R-0Ce catalyst, CH₂O^{*} formation has a high activation free energy of 2.00 eV and a reaction free energy of 0.58 eV, whereas CH₂^{*} dehydrogenation to CH^{*} is kinetically more favorable, with an activation free energy of 1.02 eV and reaction free energy of 0.49 eV. Therefore, CH^{*} formation dominates on R-0Ce, leading to a higher risk of carbon accumulation. In contrast, on the R-0.2Ce catalyst (Fig. 7c, Figs. S49-S50), CH₂^{*} more readily reacts with active oxygen to form CH₂O^{*}, with a lower activation free energy of 1.57 eV, while the competing

dehydrogenation pathway to CH^* requires 1.74 eV. This suggests that Ce doping suppresses excessive CH_2^* dehydrogenation, thus inhibiting carbon deposition and promoting CH_2O^* formation. Additionally, the oxidation of CH_2^* to CH_2O^* on R-0.2Ce is more kinetically favorable compared to R-0Ce (1.57 eV vs. 2.00 eV), further confirming the role of Ce in mitigating carbon buildup.

Line 405: As shown in Figs. 7d and 7e, the ϵ_d value of R-0.2Ce (-1.81 eV) is closer to the Fermi level than that of R-0Ce (-2.03 eV), indicating stronger CO_2 adsorption and activation on R-0.2Ce. This facilitates C-O bond cleavage and the generation of adsorbed CO and reactive oxygen species. These species can effectively react with carbon derived from CH_2^* dehydrogenation, thereby suppressing carbon accumulation. Moreover, differential charge density and Bader charge analysis (Fig. 7f, Tables S9-S10) reveal electron transfer from the CeO_{2-x} cluster to the Ni cluster, significantly enhancing the metal-support interaction. This results in stronger anchoring of exsolved Ni nanoparticles to the perovskite surface, improving their thermal stability and reducing sintering during DRM, as confirmed by SEM and TEM analyses. The electron donation also increases the d -orbital electron density of Ni, potentially boosting the adsorption and activation of both CH_4 and CO_2 .

Fig. 7 | DFT calculations. **a** (111) oriented slab model and the segregation energy defined as the energy change upon swapping Ni and Cr between the first and third layers. Free energy profiles for the reactions related to CH_2^* intermediate on the **b** R-0Ce and **c** R-0.2Ce catalysts. Density of states and d -band center for **d** R-0Ce and **e** R-0.2Ce catalyst surfaces. **f** The differential charge density of

R-0.2Ce catalyst. Cyan and yellow regions present the electrons accumulation and depletion, respectively. La, Ce, Cr, Ni, and O atoms are shown in the red, yellow, purple, green, and blue, respectively.

Line 562: Density functional theory (DFT) calculations are implemented in the Vienna ab initio Simulation Package (VASP 5.4).^[1] Perdew-Burke-Ernzerhof (PBE) with generalized gradient approximation (GGA) was used to describe exchange-correlation functional.^[2] A vacuum layer of 15 Å was introduced to avoid interactions between periodic images. A cutoff energy of 400 eV for the plane-wave basis sets and a 3×3×1 grid generated by Monkhorst-Pack method were adopted for geometric optimization calculations.^[3] Meanwhile, the k-point grid in the Brillouin zone was set to be 8×8×1 by the Monkhorst-Pack scheme when calculating the electronic structure. Structure optimizations were performed until the total energies converged to 5×10⁻⁶ eV and the forces acting on the relaxed ions were less than 0.05 eV/Å. To counteract erroneous electron delocalization, GGA+U methods were applied to La, Cr and Ni atoms. Values of U_{La}=5 eV, U_{Cr}=2 eV, U_{Ni}=3.5 eV, U_{Ce}=6 eV were utilized.^[4-7] The climbing-image nudged elastic band (CI-NEB) method^[8] was employed to search the transition states (TS), then, the dimer method was used to optimize the located TS.^[9] Moreover, the TS structure was confirmed with only one imaginary frequency.

Both R-0Ce and R-0.2Ce catalyst models were constructed based on the experimental characterization results that the (111) crystal plane was the dominantly exposed crystallographic surface. As shown in Fig. S50, The R-0Ce catalyst was modeled using a three-layer *p*(2×1) LaCrO₃(111) supercell, in which the bottom two layers were fixed. In this model, a Ni₄ cluster was located on the surface, with an adjacent oxygen vacancy surrounding the Ni₄ cluster. The R-0.2Ce model was

obtained by loading a CeO₂ cluster on the R-0Ce surface and creating an oxygen vacancy within the cluster by removing one oxygen atom, resulting in a CeO_{2-x} cluster structure. To calculate Ni segregation energy on the (111) plane, a model for Ni-doped LaCrO₃ catalyst was constructed using a five-layer $p(1\times 1)$ supercell with the bottom two layers fixed.

The adsorption free energy (G_{ads}), activation free energy (G_{a}), and reaction free energy (ΔG) were determined by the following equations:^[10]

$$G_{\text{ads}}(T, P) = E_{\text{mol/surface}} - E_{\text{mol}} - E_{\text{surface}} + \Delta U_{\text{ads}} + \Delta E_{\text{ZPE}_{\text{ads}}} - T\Delta S_{\text{ads}}$$

$$G_{\text{a}}(T, P) = E_{\text{TS}} - E_{\text{IS}} + \Delta E_{\text{ZPE}_{\text{a}}} + \Delta U_{\text{a}} - T\Delta S_{\text{a}}$$

$$\Delta G(T, P) = E_{\text{FS}} - E_{\text{IS}} + \Delta E_{\text{ZPE}} + \Delta U - T\Delta S$$

where $E_{\text{mol/surface}}$, E_{mol} , and E_{surface} were the total free energies of the adsorbed system, the clean surfaces, and the free molecules, respectively. Considering our optimal catalytic reaction conditions were 750°C and 1.0 atm, all energies were corrected at 1023.15 K and 1.0 atm using VASPKIT software.

Fig. S49 The structures of initial states and final states for CH₂ dissociation into CH* or reacting with O to CH₂O* on the (a) R-0Ce and (b) R-0.2Ce catalysts.

Fig. S50 Surface morphology of (a) R-0Ce and (b) R-0.2Ce catalysts. Red, light blue, blue-gray, steel blue, and beige balls represent O, La, Cr, Ni and Ce atoms, respectively.

Table S9 Bader charge of Ni atoms (e), as well as the total Bader charge of Ni₄ cluster on the R-0Ce and R-0.2Ce catalysts

Bader charge	R-0Ce	R-0.2Ce
Ni ₁	0.21	0.03
Ni ₂	0.35	0.19
Ni ₃	-0.47	-0.48
Ni ₄	-0.12	-0.17
Total	-0.03	-0.43

Table S10 Bader charge of Ce atoms (e), as well as the total Bader charge of CeO_{2-x} cluster on the R-0Ce and R-0.2Ce catalysts

Bader charge	CeO _{2-x}	R-0.2Ce
Ce	1.15	1.87
O	-1.04	-1.31
Total	0.11	0.56

Based on your valuable comments, we further conducted longer cyclic DRM stability tests under new working conditions and updated the performance comparison charts.

Line 96: Furthermore, the R-0.2Ce catalyst was subjected to 800 h long-term stability test under more severe operating conditions (Fig. 1e and Fig. S12). It demonstrates excellent stability, maintaining a CH₄ conversion rate of 87.4% and a CO₂ conversion rate of 92.9% after 800 h, with negligible deactivation compared to its initial performance. In contrast, the R-0Ce sample exhibits a gradually declined performance after approximately 65 h, with pronounced deactivation observed after switching to 750 °C at 80 h. This resulted in reduced CH₄ and CO₂ conversion rates of 61.7% and 73.6%, respectively.

Line 110: The R-0.2Ce catalyst exhibits higher apparent methane activity and lower carbon formation rate at same DRM temperatures, which demonstrates obvious cutting edge compared with previously reported catalysts (Fig. 1h and Table S1).

Fig. 1 | DRM Performance of R-xCe samples. CH₄, CO₂ conversion rates over various samples with different Ce doping ratios in the DRM reaction. **b** H₂, CO selectivity and H₂/CO ratio. **c** The conversion rates of CH₄ and CO₂, selectivity of H₂ and CO, and H₂/CO ratio at different temperatures for sample 0.2Ce. **d** The conversion rates of CH₄ and CO₂, selectivity of H₂ and CO, and H₂/CO value at different WHSV for sample 0.2Ce at 750 °C. **e** Stability test of R-0.2Ce and R-0Ce for 800 h and 120 h. Evaluated conditions: 800/750 °C, CH₄: CO₂: N₂=1: 1: 1, WHSV=30000

$\text{mL}\cdot\text{g}^{-1}\cdot\text{h}^{-1}$. Arrhenius plot in terms of the rate of product of f H_2 and g CO catalyzed by the R-0Ce and R-0.2Ce catalysts. 25 mg of each catalyst was loaded into the quartz tube. **h** Comparison of the DRM performance with literature (Table S1).

Fig. S12. H_2/CO value of R-0.2Ce and R-0Ce in the DRM stability test.

References

- [1] H. Wang, G. Cui, H. Lu, Z. Li, L. Wang, H. Meng, J. Li, H. Yan, Y. Yang, M. Wei, Facilitating the dry reforming of methane with interfacial synergistic catalysis in an Ir@CeO_{2-x} catalyst, *Nat Commun.*, 15 (2024) 3765.
- [2] S. Goedecker, M. Teter, J. Hutter. Separable dual-space Gaussian pseudopotentials. *Phys. Rev.*, B 54 (1996) 1703.
- [3] J. P. Perdew, K. Burke, M. Ernzerhof. Generalized gradient approximation made simple. *Phys. Rev. Lett.* 77 (1996) 3865.
- [4] H. J. Monkhorst, D. P. James. Special points for Brillouin-zone integrations. *Phys. Rev. B* 13 (1976) 5188.
- [5] L. Webster, L. Liang, J. Yan. Distinct spin-lattice and spin-phonon interactions in monolayer magnetic CrI_3 . *Phys. Chem. Chem. Phys.* 20 (2018) 23546-23555.
- [6] H. Yu, Y. Ji, C. Li, W. Zhu, Y. Wang, Z. Hu, J. Zhou, C. Pao, W. Huang, Y. Li, X. Huang, Q. Shao. Strain-triggered distinct oxygen evolution reaction pathway in two-dimensional metastable phase IrO_2 via CeO_2 loading. *J. Am. Chem. Soc.* 146 (2024) 20251-20262.
- [7] Z. Yang, T.K. Woo, K. Hermansson. Effects of Zr doping on stoichiometric and reduced ceria: A first-principles study. *J. Chem. Phys.* 124 (2006) 224704.
- [8] M. A. Preciado, A. Kassiba, A. Morales-Acevedoc, M. Makowska-Janusik. Vibrational and electronic peculiarities of NiTiO_3 nanostructures inferred from first principal calculations. *RSC Adv.* 5 (2015) 17396-17404.
- [9] G. Henkelman, J. Hanne. Improved tangent estimate in the nudged elastic band method for finding minimum energy paths and saddle points. *J. Chem. Phys.* 113 (2000) 9978-9985.
- [10] G. Henkelman, J. Hanne. A dimer method for finding saddle points on high dimensional

potential surfaces using only first derivatives. J. Chem. Phys 111 (1999) 7010-7022.

[11] J. P. Clay, J. P. Greeley, F. H. Ribeiro, W. N. Delgass, W. F. Schneider. DFT comparison of intrinsic WGS kinetics over Pd and Pt. J. Catal. 320 (2014) 106-117.

Comment 2: In addition, there are multiple sentences in the manuscript that make no sense. For example, on page 14 the sentence in line 271 starting with "As confirmed by..." The word "conductive" is wrong here.

Our response: Thank you very much for your valuable feedback. Based on your comment, the sentence "As confirmed by..." has been updated to be "As confirmed by TEM results, it is revealed that excessively high Ce doping is not conducive to exsolving Ni nanoparticles." Moreover, we also carefully checked the entire manuscript and streamlined some of the sentences.

Reviewer #2:

The paper you mentioned provides valuable insights into the effects of Ce doping on dry methane reforming. However, the stability test duration of 120 hours is significantly shorter compared to other studies, making the claims of stability questionable. Additionally, using dilute gases like nitrogen at levels exceeding 50% during DRM measurements does not seem appropriate for assessing the inherent performance of the catalyst. Several studies cited in the paper do not use dilute gases and report catalysts that maintain high performance and stability over 1000 hours. Therefore, it is recommended that experimental data obtained without inert gases be presented. To meet the standards of publication in Nature Communications, it is crucial to adequately address the comments mentioned above.

Our response: Thank you for your in-depth review of our paper. Based on your comments, 800 h long-term stability tests were conducted under $\text{CH}_4: \text{CO}_2: \text{N}_2=1: 1: 1$, $\text{WHSV}=30000 \text{ mL} \cdot \text{g}_{\text{cat}}^{-1} \cdot \text{h}^{-1}$ conditions, and the results were presented below:

Fig 1. Stability test of $0.2\text{Ce-La}_{0.97}\text{Ni}_{0.4}\text{Cr}_{0.6}\text{O}_3$ (R-0.2Ce) and $\text{LaNi}_{0.4}\text{Cr}_{0.6}\text{O}_3$ (R-0Ce) for 800 h and 120h. Evaluated conditions: 800/750 °C, $\text{CH}_4: \text{CO}_2: \text{N}_2=1: 1: 1$, $\text{WHSV}=30000 \text{ mL} \cdot \text{g}^{-1} \cdot \text{h}^{-1}$.

Furthermore, we performed a series of characterizations on the samples after the new long-term test. Including XRD, XPS, TG, Raman, SEM, and TEM. DFT theoretical calculations were also carried out to better illustrate the reasons for the long-term stability.

Detailed revisions:

Regarding the issues related to the stability of Ce-doped catalysts in dry methane reforming and the experimental conditions, we have conducted supplementary experiments and revisions. We conducted long-term tests again at 800 and 750 °C.

Line 96: Furthermore, the R-0.2Ce catalyst was subjected to 800 h long-term stability test under more severe operating conditions (Fig. 1e and Fig. S12). It demonstrates excellent stability, maintaining a CH₄ conversion rate of 87.4% and a CO₂ conversion rate of 92.9% after 800 h, with negligible deactivation compared to its initial performance. In contrast, the R-0Ce sample exhibits a gradually declined performance after approximately 65 h, with pronounced deactivation observed after switching to 750 °C at 80 h. This resulted in reduced CH₄ and CO₂ conversion rates of 61.7% and 73.6%, respectively.

Line 110: The R-0.2Ce catalyst exhibits higher apparent methane activity and lower carbon formation rate at same DRM temperatures, which demonstrates obvious cutting edge compared with previously reported catalysts (Fig. 1h and Table S1).

Fig. 1 | DRM Performance of R- x Ce samples. **a** CH_4 , CO_2 conversion rates over various samples with different Ce doping ratios in the DRM reaction. **b** H_2 , CO selectivity and H_2/CO ratio. **c** The conversion rates of CH_4 and CO_2 , selectivity of H_2 and CO , and H_2/CO ratio at different temperatures for sample 0.2Ce. **d** The conversion rates of CH_4 and CO_2 , selectivity of H_2 and CO , and H_2/CO value at different WHSV for sample 0.2Ce at 750 °C. **e** Stability test of R-0.2Ce and R-0Ce for 800 h and 120 h. Evaluated conditions: 800/750 °C, CH_4 : CO_2 : N_2 =1: 1: 1, WHSV =30000 $\text{mL}\cdot\text{g}^{-1}\cdot\text{h}^{-1}$. Arrhenius plot in terms of the rate of product of **f** H_2 and **g** CO catalyzed by the R-0Ce and R-0.2Ce catalysts. 25 mg of each catalyst was loaded into the quartz tube. **h** Comparison of the DRM performance with literature (Table S1).

Fig S12. H_2/CO value of stability test of R-0Ce and R-0.2Ce.

Line 329: Ce^{3+} exhibits strong redox capabilities that promote the dynamic capture and release of lattice oxygen, thereby facilitating the removal of surface carbon. The observed increase in Ni^0 content is attributed to the reducing environment of dry reforming of methane (DRM), which facilitates the continuous segregation of Ni ions from the perovskite bulk phase, forming a larger population of Ni^0 species. This dynamic migration contributes to the sustained enhancement in catalytic activity during the initial reaction stage, reflecting a dynamic equilibrium within the catalytic system under DRM conditions.

Line 337: Thermogravimetric analysis (Fig. 3f) revealed that the carbon deposition on R-0.2Ce after 800 h of reaction led to only a 1.6% weight loss, while R-0Ce showed a 6.4% loss after just 120 h, underscoring the superior carbon resistance of R-0.2Ce. Raman spectroscopy further confirmed the reduced carbon deposition after Ce doping (Fig. S45).

Fig. 5 | Characterizations of long-term reacted $x\text{Ce-La}_{0.97}\text{Ni}_{0.4}\text{Cr}_{0.6}\text{O}_3$ perovskite catalysts. a XRD patterns of reacted samples. XPS spectra of **b** Ce 3d. **c** O 1s. **d** Ni 3p. **e** Cr 2p. **f** TG of 120h-reacted R-0Ce and 800h-reacted R-0.2Ce.

Fig S44. XRD patterns of reacted R-0Ce samples.

Fig S45. Raman spectra of reacted R-0Ce and R-0.2Ce samples.

Line 342: SEM and TEM characterizations (Figs. S46-S47) showed that the post-reaction morphology of R-0.2Ce closely resembles that of the fresh sample, with only minimal formation of carbon nanotubes. In contrast, R-0Ce displayed significant sintering of the perovskite surface and agglomeration of Ni nanoparticles, along with more pronounced carbon nanotube formation. Quantitatively, the average Ni particle size in R-0.2Ce increased moderately from 37.9 nm to 61.8 nm after 800 h, whereas in R-0Ce it grew substantially from 69.0 nm to 116.9 nm. These results confirm that Ce doping not only mitigates carbon deposition but also improves the anti-sintering stability of the catalyst.

Fig S46. **a** SEM images of 800h-reacted R-0.2Ce. **b** Particle size of R-0.2Ce. **c** SEM images of 120h-reacted R-0 Ce. **d** Particle size of R-0Ce.

SI: The TEM image of 800h-reacted R-0.2Ce (as shown in Fig. S4) further shows that some of the nanoparticles still maintain a small particle size (<50 nm). In addition, it can be found that the distribution of Ce is mostly clustered near the exsolved Ni nanoparticles.

Fig S47. TEM images and EDS mapping of 800h-reacted R-0.2Ce.

Line 410: Moreover, differential charge density and Bader charge analysis (Fig. 7f, Tables S9-S10) reveal electron transfer from the CeO_{2-x} cluster to the Ni cluster, significantly enhancing the metal–support interaction. This results in stronger anchoring of exsolved Ni nanoparticles to the perovskite surface, improving their thermal stability and reducing sintering during DRM, as confirmed by SEM and TEM analyses. The electron donation also increases the d -orbital electron density of Ni, potentially boosting the adsorption and activation of both CH_4 and CO_2 .

Fig. 7 | DFT calculations. **a** (111) oriented slab model and the segregation energy defined as the energy change upon swapping Ni and Cr between the first and third layers. Free energy profiles for the reactions related to CH_2^* intermediate on the **b** R-0Ce and **c** R-0.2Ce catalysts. Density of states and d -band center for **d** R-0Ce and **e** R-0.2Ce catalyst surfaces. **f** The differential charge density of R-0.2Ce catalyst. Cyan and yellow regions present the electrons accumulation and depletion, respectively. La, Ce, Cr, Ni, and O atoms are shown in the red, yellow, purple, green, and blue, respectively.

Table S9 Bader charge of Ni atoms (e), as well as the total Bader charge of Ni_4 cluster on the R-0Ce and R-0.2Ce catalysts.

Bader charge	R-0Ce	R-0.2Ce
Ni ₁	0.21	0.03
Ni ₂	0.35	0.19
Ni ₃	-0.47	-0.48
Ni ₄	-0.12	-0.17

Total	-0.03	-0.43
-------	-------	-------

Table S10 Bader charge of Ce atoms (e), as well as the total Bader charge of CeO_{2-x} cluster on the R-0Ce and R-0.2Ce catalysts.

Bader charge	CeO_{2-x}	R-0.2Ce
Ce	1.15	1.87
O	-1.04	-1.31
Total	0.11	0.56

Reviewer #3:

This is a very strong paper with significant novelty and impact. The thorough characterization, mechanistic insights, and excellent performance of the catalyst make it a valuable contribution to the field of dry reforming of methane. The weaknesses are relatively minor and can be addressed with some clarifications and a slightly expanded discussion in certain areas. I would highly recommend this paper for publication after minor revisions addressing the points below.

Our response:

Thank you very much for your positive and encouraging comments regarding our manuscript. We have expanded the discussion in areas where further elaboration was needed. In particular, we have included additional details on the mechanistic aspects of

the catalyst's performance and provided a broader context by comparing our findings with recent literature. Based on your comments and useful advice, the revisions we have made are as follows:

Specific responses:

Comment 1: While the paper mentions carbon deposition and shows evidence of MWCNT formation, a more in-depth discussion of the type of carbon formed (beyond Raman ID/IG ratios) would be beneficial. TEM images after the long-term stability test are shown in the SI, showing MWCNTs. Elaborating on the mechanism of carbon resistance (why the MWCNTs don't lead to rapid deactivation in the R-0.2Ce sample) in the main text would strengthen the conclusions. The connection between the increased $\text{Ce}^{3+}\text{-O}_v$ structures after reaction (Fig 5b, c) and carbon resistance could be explicitly stated.

Our response: Thank you very much for your valuable feedback. Based on your comment 4, we have conducted long-term tests again at 800 °C and 750 °C. We have carried out the relevant characterization of the post-reaction samples under the new test conditions, showing that the amount of R-0.2Ce carbon build-up is only 1.6 wt% after 800 h. We have also analyzed the TEM images of the post-reaction samples, which together with the DFT calculations have demonstrated the role of CeO_{2-x} in the resistance to carbon deposition and sintering agglomeration (due to the trace amount of carbon deposition under the new test conditions we have weakened the analysis of the carbon build-up portion of the paper). In addition, the post-reaction XPS was analyzed and the link between the increased $\text{Ce}^{3+}\text{-O}_v$ structure and the anti-carbon accumulation was clearly indicated in the main text.

Detailed revisions:

Line 96: Furthermore, the R-0.2Ce catalyst was subjected to 800 h long-term stability test under more severe operating conditions (Fig. 1e and Fig. S12). It demonstrates excellent stability, maintaining a CH_4 conversion rate of 87.4% and a CO_2 conversion rate of 92.9% after 800 h, with negligible deactivation compared to its initial performance. In contrast, the R-0Ce sample exhibits a gradually declined performance

after approximately 65 h, with pronounced deactivation observed after switching to 750 °C at 80 h. This resulted in reduced CH₄ and CO₂ conversion rates of 61.7% and 73.6%, respectively.

Line 110: The R-0.2Ce catalyst exhibits higher apparent methane activity and lower carbon formation rate at same DRM temperatures, which demonstrates obvious cutting edge compared with previously reported catalysts (Fig. 1h and Table S1).

Fig. 1 | DRM Performance of R-xCe samples. **a** CH₄, CO₂ conversion rates over various samples with different Ce doping ratios in the DRM reaction. **b** H₂, CO selectivity and H₂/CO ratio. **c** The conversion rates of CH₄ and CO₂, selectivity of H₂ and CO, and H₂/CO ratio at different temperatures for sample 0.2Ce. **d** The conversion rates of CH₄ and CO₂, selectivity of H₂ and CO, and H₂/CO value at different WHSV for sample 0.2Ce at 750 °C. **e** Stability test of R-0.2Ce and R-0Ce for 800 h and 120 h. Evaluated conditions: 800/750 °C, CH₄: CO₂: N₂=1: 1: 1, WHSV=30000 mL·g⁻¹·h⁻¹. Arrhenius plot in terms of the rate of product of **f** H₂ and **g** CO catalyzed by the R-0Ce and R-0.2Ce catalysts. 25 mg of each catalyst was loaded into the quartz tube. **h** Comparison of the DRM performance with literature (Table S1).

Line 342: Thermogravimetric analysis (Fig. 3f) revealed that the carbon deposition on R-0.2Ce after 800 h of reaction led to only a 1.6% weight loss, while R-

0Ce showed a 6.4% loss after just 120 h, underscoring the superior carbon resistance of R-0.2Ce. Raman spectroscopy further confirmed the reduced carbon deposition after Ce doping (Fig. S45).

Fig. 5 | Characterizations of long-term reacted $x\text{Ce-La}_{0.97}\text{Ni}_{0.4}\text{Cr}_{0.6}\text{O}_3$ perovskite catalysts. a XRD patterns of reacted samples. XPS spectra of **b** Ce 3d. **c** O 1s. **d** Ni 3p. **e** Cr 2p. **f** TG of 120h-reacted R-0Ce and 800h-reacted R-0.2Ce.

Figure S45. Raman spectra of 120h-reacted R-0Ce and 800h-reacted R-0.2Ce.

Line 342: SEM and TEM characterizations (Figs. S46-S47) showed that the post-reaction morphology of R-0.2Ce closely resembles that of the fresh sample, with only minimal formation of carbon nanotubes. In contrast, R-0Ce displayed significant sintering of the perovskite surface and agglomeration of Ni nanoparticles, along with more pronounced carbon nanotube formation. Quantitatively, the average Ni particle

size in R-0.2Ce increased moderately from 37.9 nm to 61.8 nm after 800 h, whereas in R-0Ce it grew substantially from 69.0 nm to 116.9 nm. These results confirm that Ce doping not only mitigates carbon deposition but also improves the anti-sintering stability of the catalyst.

Fig S46. **a** SEM images of 800h-reacted R-0.2Ce. **b** Particle size of R-0.2Ce. **c** SEM images of 120h-reacted R-0 Ce. **d** Particle size of R-0Ce.

SI: The TEM image of 800h-reacted R-0.2Ce (as shown in Fig. S4) further shows that some of the nanoparticles still maintain a small particle size (<50 nm). In addition, it can be found that the distribution of Ce is mostly clustered near the exsolved Ni nanoparticles.

Fig S47. TEM images and EDS mapping of 800h-reacted R-0.2Ce.

Line 329: Ce^{3+} exhibits strong redox capabilities that promote the dynamic capture and release of lattice oxygen, thereby facilitating the removal of surface carbon. The observed increase in Ni^0 content is attributed to the reducing environment of dry reforming of methane (DRM), which facilitates the continuous segregation of Ni ions from the perovskite bulk phase, forming a larger population of Ni^0 species. This dynamic migration contributes to the sustained enhancement in catalytic activity during the initial reaction stage, reflecting a dynamic equilibrium within the catalytic system under DRM conditions.

Comment 2: While Figure 1h and Table S1 compare the DRM performance to previous research, a more direct comparison in the discussion section to similar perovskite-based catalysts (specifically those with Ni and/or Ce) would be helpful. This would more clearly highlight the advantages of the current catalyst design. Are there other examples of dual-role Ce doping in perovskites, and how does this work compare?

Our response: Thank you very much for your comment. Based on your suggestion, we have redrawn the performance comparison graph according to the new DRM test results. It can be seen that our designed catalytic system has a combined advantage in both methane specific activity and carbon formation rate, and the reaction time is significantly higher than the current perovskite catalytic system.

Fig 1h. Comparison of the DRM performance with literature (Table S1).

Detailed revisions:

Line 110: The R-0.2Ce catalyst exhibits higher apparent methane activity and lower carbon formation rate at same DRM temperatures, which demonstrates obvious cutting edge compared with previously reported catalysts (Fig. 1h and Table S1).

Table S1. Comparison of the DRM performance with literature.

Catalysts	Active metals loading (wt %)	Reaction conditions	Specific activity ($\text{mmol}\cdot\text{g}_{\text{active metals}}^{-1}\cdot\text{s}^{-1}$) / Conversion (%)		H_2/CO	GHSV ($\text{mL}\cdot\text{g}_{\text{cat}}^{-1}\cdot\text{h}^{-1}$)	T ($^{\circ}\text{C}$)	TOS (h)	Coke formation rate ($\text{mmol}\cdot\text{g}_{\text{ca}}^{-1}\cdot\text{s}^{-1}$)	Ref.
			CH ₄	CO ₂						
0.2Ce-La_{0.97}Ni_{0.4}Cr_{0.6}O₃	7.76% Ni	CH₄:CO₂:N₂=1:1:1	1.38 / 87.4	1.49 / 92.9	0.92	30000	800	800	4.63E-07	This work
LaNi_{0.4}Cr_{0.6}O₃	8.55% Ni	CH₄:CO₂:N₂=1:1:1	1.16 / 79.7	1.27 / 87.9	0.89	30000	800	120	1.23E-05	This work
La(Co _{0.1} Ni _{0.9}) _{0.5} Fe _{0.5} O ₃	12.03% Ni-Co	CH ₄ :CO ₂ =1:1	0.43 / 70.0	0.49 / 80.2	0.87	12000	750	30	6.17E-07	[7]
La(Co _{0.3} Ni _{0.7}) _{0.5} Fe _{0.5} O ₃	12.02% Ni-Co	CH ₄ :CO ₂ =1:1	0.43 / 70.2	0.50 / 80.0	0.88	12000	750	30	1.16E-06	[7]
La _{0.9} Sr _{0.1} Ni _{0.5} Fe _{0.5} O ₃	12.60% Ni	CH ₄ :CO ₂ :He=1:1:1	0.27 / 47.0	0.40 / 55.0	/	18000	700	50	1.85E-05	[8]

$\text{La}_{0.9}\text{Sr}_{0.1}\text{NiO}_3$	23.60% Ni	$\text{CH}_4:\text{CO}_2:\text{He}=1:1:1$	0.22 / 69.9	0.26 / 70.0	/	18000	700	8	8.97E-04	[8]
$\text{LaNi}_{0.6}\text{Mn}_{0.4}\text{O}_3$	14.46% Ni	$\text{CH}_4:\text{CO}_2:\text{N}_2=1:1:2$	0.30 / 91.0	0.45 / 85.0	1.02	15000	750	10	7.16E-05	[9]
LaNiO_3	23.96% Ni	$\text{CH}_4:\text{CO}_2:\text{N}_2=1:1:2$	0.16 / 75.0	0.23 / 70.0	1.08	15000	750	10	3.77E-04	[9]
$\text{LaFe}_{0.8}\text{Ni}_{0.2}\text{O}_3^{\text{a}}$	4.82% Ni	$\text{CH}_4:\text{CO}_2:\text{He}=1:1:18$	0.35 / 75.0	0.39 / 85.0	0.78	36000	700	24	/	[10]
$\text{LaFe}_{0.8}\text{Ni}_{0.2}\text{O}_3^{\text{b}}$	4.82% Ni	$\text{CH}_4:\text{CO}_2:\text{He}=1:1:18$	0.32 / 70.0	0.38 / 82.0	0.73	36000	700	24	/	[10]
$\text{PrBaFeCoO}_{5+\delta}$	23.47% Co-Fe	$\text{CH}_4:\text{CO}_2:\text{N}_2=1:1:3$	0.04 / 13.0	0.09 / 28.0	0.35	30000	900	120	6.94E-06	[11]
$\text{La}_{0.9}\text{Ca}_{0.1}\text{Ni}_{0.5}\text{Fe}_{0.5}\text{O}_3$	12.53% Ni	$\text{CH}_4:\text{CO}_2=1:1$	1.06 / 72.0	1.14 / 77.0	/	30000	750	500	≈ 0	[12]
$\text{La}_{0.9}\text{Ca}_{0.1}\text{Ni}_{0.3}\text{Fe}_{0.7}\text{O}_3$	7.54% Ni	$\text{CH}_4:\text{CO}_2=1:1$	0.44 / 18.0	1.01 / 41.0	/	30000	750	130	8.90E-04	[12]
$\text{La}_{0.9}\text{Ca}_{0.1}\text{NiO}_3$	24.90% Ni	$\text{CH}_4:\text{CO}_2=1:1$	0.31 / 42.0	0.49 / 66.0	/	30000	750	40	/	[12]
$\text{Pr}_{0.5}\text{Ba}_{0.5}\text{Mn}_{0.8}\text{Ni}_{0.15}\text{Fe}_{0.05}\text{O}_3$	4.78% Ni-Fe	$\text{CH}_4:\text{CO}_2:\text{N}_2=33:34:3$ 3	1.34 / 52.0	1.53 / 58.0	0.80	30000	800	260	4.71E-08	[13]

$\text{Pr}_{0.5}\text{Ba}_{0.5}\text{Mn}_{0.8}\text{Ni}_{0.2}\text{O}_3$	4.83% Ni	$\text{CH}_4:\text{CO}_2:\text{N}_2=33:34:3$ 3	2.03 / 80.0	2.12 / 81.0	0.84	30000	800	135	1.03E-04	[13]
$\text{La}_{0.9}\text{Sr}_{0.1}\text{Fe}_{0.95}\text{Ni}_{0.05}\text{O}_3$	1.23% Ni	$\text{CH}_4:\text{CO}_2:\text{Ar}=1.5:1:2$ 7.5	0.28 / 93.0	0.20 / 99.0	1.10	6000	900	20	1.74E-04	[14]
$\text{Pr}_{0.45}\text{Ba}_{0.45}\text{Mn}_{0.8}(\text{Co}_{1/3}\text{Ni}_{2/3})_{0.2}\text{O}_{3\pm\delta}$	5.14% Ni-Co	$\text{CH}_4:\text{CO}_2:\text{N}_2=1:1:8$	0.17 / 84.0	0.19 / 93.0	0.93	8570	850	50	/	[15]
$\text{Pr}_{0.5}\text{Ba}_{0.5}\text{Mn}_{0.85}\text{Ni}_{0.1}\text{Fe}_{0.045}\text{Rh}_{0.05}\text{O}_3$	3.67% Ni-Fe-Rh	$\text{CH}_4:\text{CO}_2:\text{N}_2=33:34:3$ 3	2.80 / 70.0	3.35 / 81.0	0.84	36000	750	120	3.87E-07	[16]
$\text{La}_{0.8}\text{Sm}_{0.2}\text{NiO}_{3-\delta}$	23.67% Ni	$\text{CH}_4:\text{CO}_2:\text{Ar}=3:2:5$	0.25 / 48.0	0.42 / 81.0	/	40000	750	18	/	[17]
$\text{La}_{0.8}\text{Ce}_{0.2}\text{NiO}_{3-\delta}$	23.87% Ni	$\text{CH}_4:\text{CO}_2:\text{Ar}=3:2:5$	0.19 / 37.0	0.33 / 64.0	/	40000	750	18	/	[17]
LaNiO_3	23.96% Ni	$\text{CH}_4:\text{CO}_2:\text{N}_2:\text{He} =$ 1.41:0.93:0.18:1	0.30 / 95.0	0.19 / 95.0	1.75	15000	850	50	1.39E-04	[18]
$\text{La}_2\text{Ni}_{0.8}\text{Cu}_{0.2}\text{O}_4$	14.87% Ni-Cu	$\text{CH}_4:\text{CO}_2:\text{N}_2=1:1:2$	0.22 / 88.0	0.23 / 93.0	0.91	12000	750	50	2.22E-05	[19]
$\text{La}_2\text{Ni}_{0.6}\text{Cu}_{0.4}\text{O}_4$	15.06% Ni-Cu	$\text{CH}_4:\text{CO}_2:\text{N}_2=1:1:2$	0.21 / 85.0	0.22 / 90.0	0.96	12000	750	50	2.23E-04	[19]
$\text{LaFe}_{0.7}\text{Ni}_{0.1}\text{Co}_{0.1}\text{Cu}_{0.05}\text{Pd}_{0.05}\text{O}_3$	8.23% Ni-Co- Cu-Pd	$\text{CH}_4:\text{CO}_2:\text{N}_2=1:1:18$	0.06 / 70.0	0.06 / 79.0	/	/	700	24	/	[20]

LaFeNi _{0.05} Co _{0.05} O ₃	2.37% Ni-Co	CH ₄ :CO ₂ :N ₂ =1:1:18	0.22 / 78.0	0.21 / 85.0	/	/	800	24	/	[20]
La _{1.8} Ba _{0.2} Ni _{0.9} Cu _{0.1} O ₄	14.77% Ni-Cu	CH ₄ :CO ₂ :He=1:1:3	0.93 / 46.0	1.21 / 60.0	0.90	120000	700	6	/	[21]
PrBaMn _{1.7} Co _{0.1} Ni _{0.2} O _{5+δ} + 15 wt % infiltration of Fe	18.20% Ni-Co-Fe	CH ₄ :CO ₂ :He=1:1:3	0.03	0.06	/	30000	750	350	≈ 0	[22]
La _{0.6} Sr _{0.2} Cr _{0.85} Ni _{0.15} O ₃	4.37% Ni	CH ₄ :CO ₂ :N ₂ =9:9:2	0.34 / 88.3	0.30 / 88.0	/	3000	750	24	≈ 0	[23]
La _{0.8} Ca _{0.2} Mn _{0.8} Ni _{0.2} O _{3±δ}	6.19% Ni	CH ₄ :CO ₂ :Ar=1:1:3	0.63 / 88.2	0.69 / 96.6	/	18000	800	500	3.47E-07	[24]
La ₂ Ni _{0.6} Cu _{0.4} O ₄	8.75% Ni 6.32% Cu	CH ₄ :CO ₂ :N ₂ =1:1:2	0.21 / 88.0	0.23 / 92.0	/	12000	750	50	2.22E-05	[25]
Ba _{0.9} Zr _{0.5} Ce _{0.2} Y _{0.2} Ni _{0.1} O _{3-δ}	2.18% Ni	CH ₄ :CO ₂ =1:1	1.79 / 71.0	/	/	9000	700	60	1.93E-06	[26]
La ₂ Ce ₂ O ₇ -LaNiO ₃	5.00% Ni	CH ₄ :CO ₂ =1:1	1.96 / 89.0	2.03 / 91.0	0.63	18000	850	6	7.72E-05	[27]

a: 120 min reduction

b: 30 min reduction

* Coke formation rate calculated by the equation: w (weight loss%)/(12*3.6*TOS(h))

Comment 3: The authors correctly note that the reduction temp can influence the final product. While the supplementary material explores this, a brief discussion of the optimized reduction temp (900°C) in the main body is warranted. Just a sentence or two to guide the readers to the related figures.

Our response: Thank you very much for your valuable feedback. We concur with your viewpoint that it is necessary to describe the optimized selection of reduction temperature in the main body. We investigated the effects of different reduction temperatures on the morphology and performance of the catalysts, identifying 900 °C as the optimal reduction temperature.

Detailed revisions:

Line 164: The DRM test further confirmed that the sample reduced at 900 °C exhibited the highest catalytic performance, attributed to its smaller particle sizes and higher dispersion (Fig. S23-S25). Specifically, this sample achieved the highest particle density of approximately 9.3 NPs/ μm^2 , with an average particle diameter of 37.9 nm.

Comment 4: The long-term stability test used a WHSV of 12000 $\text{mL}\cdot\text{g}^{-1}\cdot\text{h}^{-1}$. How does this compare to the space velocities where the performance remained constant? Does this low value contribute to high performance?

Our response: Thank you very much for your valuable feedback. We compared the space velocities from perovskite-catalyzed DRM tests in recent years, as shown in Fig. 1. We updated the DRM stability test at WHSV of 30,000 $\text{mL}\cdot\text{g}^{-1}\cdot\text{h}^{-1}$, CH_4 : CO_2 : $\text{N}_2=1$: 1: 1.

Table 1. Comparison of the WHSV with literature.

Sample	WHSV ($\text{L}\cdot\text{gcat}^{-1}\cdot\text{h}^{-1}$)	WHSV- CH_4 ($\text{L}\cdot\text{gcat}^{-1}\cdot\text{h}^{-1}$)
This work	30000	10000
$\text{La}(\text{Co}_{0.1}\text{Ni}_{0.9})_{0.5}\text{Fe}_{0.5}\text{O}_3$	12000	6000
$\text{La}_{0.9}\text{Sr}_{0.1}\text{Ni}_{0.5}\text{Fe}_{0.5}\text{O}_3$	18000	6000

$\text{La}_{0.9}\text{Sr}_{0.1}\text{NiO}_3$	18000	6000
$\text{LaNi}_{0.6}\text{Mn}_{0.4}\text{O}_3$	15000	3750
LaNiO_3	15000	3750
$\text{LaFe}_{0.8}\text{Ni}_{0.2}\text{O}_3^a$	36000	1800
$\text{PrBaFeCoO}_{5+\delta}$	30000	7500
$\text{La}_{0.9}\text{Ca}_{0.1}\text{Ni}_{0.5}\text{Fe}_{0.5}\text{O}_3$	30000	15000
$\text{La}_{0.9}\text{Ca}_{0.1}\text{Ni}_{0.3}\text{Fe}_{0.7}\text{O}_3$	30000	15000
$\text{Pr}_{0.5}\text{Ba}_{0.5}\text{Mn}_{0.8}\text{Ni}_{0.15}\text{Fe}_{0.05}\text{O}_3$	30000	10000
$\text{La}_{0.9}\text{Sr}_{0.1}\text{Fe}_{0.95}\text{Ni}_{0.05}\text{O}_3$	6000	300
$\text{Pr}_{0.45}\text{Ba}_{0.45}\text{Mn}_{0.8}(\text{Co}_{1/3}\text{Ni}_{2/3})_{0.2}\text{O}_{3\pm\delta}$	8570	857
$\text{Pr}_{0.5}\text{Ba}_{0.5}\text{Mn}_{0.85}\text{Ni}_{0.1}\text{Fe}_{0.045}\text{Rh}_{0.005}\text{O}_3$	36000	12000
$\text{La}_{0.8}\text{Sm}_{0.2}\text{NiO}_{3-\delta}$	40000	12000
LaNiO_3	15000	6008
$\text{La}_2\text{Ni}_{0.8}\text{Cu}_{0.2}\text{O}_4$	12000	3000
$\text{La}_{1.8}\text{Ba}_{0.2}\text{Ni}_{0.9}\text{Cu}_{0.1}\text{O}_4$	120000	24000
$\text{PrBaMn}_{1.7}\text{Co}_{0.1}\text{Ni}_{0.2}\text{O}_{5+\delta}$	30000	6000
+ 15 wt % infiltration of Fe		
$\text{La}_{0.6}\text{Sr}_{0.2}\text{Cr}_{0.85}\text{Ni}_{0.15}\text{O}_3$	3000	1350
$\text{La}_{0.8}\text{Ca}_{0.2}\text{Mn}_{0.8}\text{Ni}_{0.2}\text{O}_{3\pm\delta}$	18000	3600
$\text{La}_2\text{Ni}_{0.6}\text{Cu}_{0.4}\text{O}_4$	12000	3000
$\text{Ba}_{0.9}\text{Zr}_{0.5}\text{Ce}_{0.2}\text{Y}_{0.2}\text{Ni}_{0.1}\text{O}_{3-\delta}$	9000	4500
$\text{La}_2\text{Ce}_2\text{O}_7\text{-LaNiO}_3$	18000	9000

Comment 5: While the H_2/CO ratio approaches the theoretical value of 1, it's consistently slightly below 1. A brief discussion of potential reasons for this (e.g., slight contribution of rWGS, even if minimized) would add completeness.

Our response: Thank you very much for your valuable comment. According to the Comment 4, we re-performed the stability tests and the H₂/CO results are shown in Fig. S12. And a brief discussion of potential reasons for the side reaction has been added.

Detailed revisions:

SI: It can be seen (Fig. S12) that the H₂/CO ratio of R-0.2Ce stays around 0.93 (800 °C) after 800 h, and the H₂/CO ratio is 0.88 (750 °C) at 240 h. For comparison, the H₂/CO value of the R-0Ce after 120 h is only 0.75 (750 °C). This result indicates that Ce modification hinders the side reactions of DRM process.

Line 83: This phenomenon can be explained by the occurrence of reverse water-gas shift (rWGS, H₂+CO₂→CO+H₂O, ΔH=+41 kJ/mol) equilibrium, and Ce doping modulates its dynamic equilibrium as well as actuating the H utilization efficiency (Figs. S4-S5).

Fig S12. H₂/CO value of stability test of R-0Ce and R-0.2Ce.

Comment 6: The explanation for the decreased performance with excessive Ce (R-0.4Ce) could be strengthened. While the authors mention coverage of Ni active sites by CeO_{2-x}, explicitly connecting this to the XPS and TEM results (showing larger Ni particle size and potentially less exposed Ni) would be beneficial.

Our response: Thank you very much for your valuable feedback. In the R-0.4Ce sample, excess CeO_{2-x} aggregated on the surface of perovskite, which hindered the segregation of Ni, resulting in a significant increase in particle size and a decrease in

the exposure sites of Ni. Meanwhile, the dual effects of physical masking and chemical oxidation of excess CeO_{2-x} inhibited the exsolution and reduction of Ni, which ultimately led to a significant decrease in the catalytic performance. Below, we present our detailed responses to your specific questions and suggestions as well as our detailed revisions.

Detailed revisions:

Line 150: Excessive Ce modification in the R-0.4Ce sample leads to CeO_{2-x} aggregation on the perovskite surface, which hinders Ni segregation and aggregates the nanoparticles on the surface, resulting in a significant decrease in the density of active sites and increase in the particle size.

Line 266: The surface Ce^{3+} content in the R-0.4Ce sample decreased to 8.5%, likely due to the aggregation of surface CeO_{2-x} species at higher Ce concentrations, which favors the formation of more stable Ce^{4+} species. Additionally, the XPS signal intensity of metallic Ni^0 was significantly reduced in the R-0.4Ce sample, while the Ni^{2+} peak was notably enhanced. These results suggest that the excess CeO_{2-x} phase promotes the chemical oxidation of surface Ni meanwhile physically hinders the exposure of metallic Ni^0 through surface coverage.

Comment 7: The TG data (Fig. S44) shows more carbon deposition on R-0.2Ce, but this sample is more stable. The paper mentions a lower graphitization degree, but a clearer, more concise statement connecting this to the overall stability is needed.

Our response: Thank you very much for your valuable comment. Based on your comments, we compared the carbon generation rate of R-0Ce and R-0.2Ce sample. Under the new test conditions, we reconfirmed the experimental results that R-0.2Ce has less carbon deposition after 800 h of reaction than R-0Ce after 120h of reaction (This may be due to changes in reaction temperature, airspeed and reactant concentration). Specific information is as follows:

Detailed revisions:

Line 337: Thermogravimetric analysis (Fig. 3f) revealed that the carbon deposition on R-0.2Ce after 800 h of reaction led to only a 1.6% weight loss, while R-0Ce showed a 6.4% loss after just 120 h, underscoring the superior carbon resistance of R-0.2Ce. Raman spectroscopy further confirmed the reduced carbon deposition after Ce doping (Fig. S45).

Line 342: SEM and TEM characterizations (Figs. S46-S47) showed that the post-reaction morphology of R-0.2Ce closely resembles that of the fresh sample, with only minimal formation of carbon nanotubes. In contrast, R-0Ce displayed significant sintering of the perovskite surface and agglomeration of Ni nanoparticles, along with more pronounced carbon nanotube formation. Quantitatively, the average Ni particle size in R-0.2Ce increased moderately from 37.9 nm to 61.8 nm after 800 h, whereas in R-0Ce it grew substantially from 69.0 nm to 116.9 nm. These results confirm that Ce doping not only mitigates carbon deposition but also improves the anti-sintering stability of the catalyst.

Fig. 5 | Characterizations of long-term reacted $x\text{Ce-La}_{0.97}\text{Ni}_{0.4}\text{Cr}_{0.6}\text{O}_3$ perovskite catalysts. a XRD patterns of reacted samples. XPS spectra of b Ce 3d. c O 1s. d Ni 3p. e Cr 2p. f TG of 120h-reacted R-0Ce and 800h-reacted R-0.2Ce.

Fig S45. Raman spectra of reacted R-0Ce and R-0.2Ce samples.

Fig S46. **a** SEM images of 800h-reacted R-0.2Ce. **b** Particle size of R-0.2Ce. **c** SEM images of 120h-reacted R-0 Ce. **d** Particle size of R-0Ce.

Fig S47. TEM images and EDS mapping of 800h-reacted R-0.2Ce.

Minor Comment 1: While generally very well-written, there are a few minor grammatical errors. A final proofread is recommended.

Our response: Thank you very much for your valuable feedback. We have thoroughly reviewed the grammar of the entire text to ensure that there is no ambiguity and that the expression is clear.

Minor Comment 2: It would improve the readability if the authors had listed all the abbreviations at the end of the paper.

Our response: Thank you very much for your comment. We also agree that it is necessary to add a list of the meanings represented by the abbreviations at Supporting Information. Below, we present the detailed version.

Detailed revisions:

Full name	Abbreviations
Calcined $\text{LaNi}_{0.4}\text{Cr}_{0.6}\text{O}_3$	C-0Ce
Calcined 0.1Ce- $\text{La}_{0.97}\text{Ni}_{0.4}\text{Cr}_{0.6}\text{O}_3$	C-0.1Ce
Calcined 0.2Ce- $\text{La}_{0.97}\text{Ni}_{0.4}\text{Cr}_{0.6}\text{O}_3$	C-0.2Ce
Calcined 0.3Ce- $\text{La}_{0.97}\text{Ni}_{0.4}\text{Cr}_{0.6}\text{O}_3$	C-0.3Ce
Calcined 0.4Ce- $\text{La}_{0.97}\text{Ni}_{0.4}\text{Cr}_{0.6}\text{O}_3$	C-0.4Ce
Reduced $\text{LaNi}_{0.4}\text{Cr}_{0.6}\text{O}_3$	R-0Ce
Reduced 0.1Ce- $\text{La}_{0.97}\text{Ni}_{0.4}\text{Cr}_{0.6}\text{O}_3$	R-0.1Ce
Reduced 0.2Ce- $\text{La}_{0.97}\text{Ni}_{0.4}\text{Cr}_{0.6}\text{O}_3$	R-0.2Ce
Reduced 0.3Ce- $\text{La}_{0.97}\text{Ni}_{0.4}\text{Cr}_{0.6}\text{O}_3$	R-0.3Ce
Reduced 0.4Ce- $\text{La}_{0.97}\text{Ni}_{0.4}\text{Cr}_{0.6}\text{O}_3$	R-0.4Ce
Dry reforming of methane	DRM
Reverse water-gas shift	rWGS
Weight hourly space velocity	WHSV
Inductively coupled plasma optical emission spectroscopy	ICP-OES
X-ray diffractometer	XRD

Scanning electron microscopy	SEM
Energy dispersive spectroscopy	EDS
transmission electron microscopy	TEM
High-angle annular dark-field scanning transmission electron microscopy	HAADF-STEM
Atomic force microscope	AFM
X-ray photoelectron spectroscopy	XPS
Electron paramagnetic resonance	EPR
In-situ diffuse reflectance infrared fourier transform spectroscopy	In-situ DRIFTS
The X-ray absorption fine structure spectra	XAS
Density functional theory	DFT
The temperature-programmed reaction of methane	CH ₄ -TPR